# PolySUMOylation of PCNA and Rad52 restricts centromeric recombination in fission yeast

Katarzyna Markowska [1], Ireneusz Litwin[1], Dorota Misiorna [1,2,6], Julia Kończak [1,3,6], Aleksandra Bogdańska [1], Paulina Tomaszewska [3,4], Michał Tracz [2,5], Mika Haenen[1,3] & Karol Kramarz [1] ✉

SUMOylation, a conserved post-translational modification in eukaryotes, regulates protein function, localization, and stability. However, the role of SUMO chains in genome maintenance is still emerging. Using *Schizosaccharomyces pombe*, we show that loss of SUMO chains results in spontaneous replication stress, DNA damage, and elevated centromeric recombination. To investigate SUMO-dependent interactome at the sites of Rad52 repair, we used a split-SUMO-ID proteomics approach. It allows the analysis of local SUMOylation content at the Rad52 repair sites, and enabled the identification of the essential replication factor PCNA. We found that SUMO chain-modified PCNA antagonizes Rad8-mediated PCNA polyubiquitination, modulating the choice of post-replication repair pathways at stalled forks within centromeres. In the absence of polySUMOylation, excessive PCNA polyubiquitination drives elevated recombination at centromeres. Artificial tethering of a SUMO chain to Rad52 suppresses this defect. Our findings uncover an essential role for SUMO chains in centromere maintenance by modulating DNA repair pathway choice under endogenous replication stress.

Eukaryotic genomes are threatened by numerous factors of endogenous and exogenous sources. Even under unchallenged conditions, replicating genomes are prone to mutations as a result of replication fork (RF) stalling on a plethora of endogenous obstacles, difficult-to-replicate regions, or collisions with transcription machinery, collectively referred to as replication stress (RS)[1]. RF arrest can lead to the formation of single-stranded DNA (ssDNA) gaps and even double-stranded breaks (DSBs) upon dissociation of replisome components[2]. In human cells, RS has emerged as a central determinant of genomic instability during the early stage of tumorigenesis[3]. Salvage pathways that enable the completion of distorted replication are collectively known as the replication stress response (RSR)[4]. Among these,

homologous recombination (HR) is a key mechanism ensuring faithful genome duplication. Canonical HR involves binding of Rad51 recombinase to ssDNA, homology search, and resolution of branched DNA structures. In yeast, the Rad52 mediator removes replication protein A (RPA), which protects ssDNA, to facilitate Rad51 loading. To avoid uncontrolled endonucleolytic digestion of collapsed forks and their conversion into DSBs, cells stabilize and protect stalled forks in a Rad51-dependent manner[5]. Subsequently, either a merger with late-fired converging forks occurs, also requiring Rad51 fork protection, or arrested forks are actively restarted by HR mechanisms[6]. The choice of the appropriate repair pathway at stressed RFs may be regulated by various post-translational modifications (PTMs)[7]. In particular,

[1]Academic Excellence Hub - Research Centre for DNA Repair and Replication, Faculty of Biological Sciences, University of Wrocław, Wrocław, Poland. [2]Faculty of Biotechnology, University of Wrocław, Wrocław, Poland. [3]Department of Genetics and Cell Physiology, Faculty of Biological Sciences, University of Wrocław, Wrocław, Poland. [4]Department of Genetics, Genomics and Cancer Sciences, University of Leicester, Leicester, UK. [5]Laboratory of Mass Spectrometry, Faculty of Biotechnology, University of Wrocław, Wrocław, Poland. [6]These authors contributed equally: Dorota Misiorna, Julia Kończak. ✉e-mail: karol.kramarz@uwr.edu.pl

modifications of the essential DNA replication factor—proliferating cell nuclear antigen (PCNA) at K164 with ubiquitin have been shown in budding yeast to channel DNA repair into post-replication repair (PRR), whereas K164 SUMOylation inhibited HR[8]. The PRR pathway comprises two branches: error-prone translesion synthesis (TLS) and error-free template switching (TS), which may involve Rad51 activity. In chicken DT40 cells, PCNA SUMOylation at K164 has been also implicated in the release of replication blocks by error-free TS[9]. Altogether, HR is a potent anti-tumor mechanism essential for preserving genome stability in humans[10].

SUMO (small ubiquitin-like modifier) is one of the key PTM that regulates metabolic processes and is crucial for cell survival. However, unlike ubiquitin, SUMO is not an abundant modification. SUMOylation, the covalent attachment of SUMO to target proteins, controls gene expression, DNA replication, cell cycle progression, and genome stability[11]. Importantly, SUMO levels are often upregulated in many types of cancer, and SUMO inhibition can reduce cancer cell proliferation[12]. However, inhibition of SUMOylation also causes toxicity in normal cells, as mutations in SUMO pathway enzymes often lead to synthetic lethality or disease, from yeast models to mice[13,14]. Thus, a detailed understanding of SUMO biology is essential. SUMOylation affects the activity, localization, or stability of modified proteins[15]. Similar to ubiquitin, SUMO is transferred to substrates through an enzymatic cascade involving E1 activating enzyme, E2 conjugating enzyme, and E3 SUMO ligases, which enhance the specificity of SUMO attachment to target proteins[16]. In unicellular fission yeast, SUMO is encoded by a single gene (pmt3+), whereas in humans, five SUMO paralogs (SUMO1–5) are expressed[17,18]. Nearly three decades of studies using yeast and other models suggested that monoSUMOylation promotes the repair of damaged DNA, and multiple proteins, including HR factors, can undergo simultaneous SUMOylation, referred to as a SUMO clouds[13,19]. Another feature of SUMO, similar to ubiquitin, is its ability to form polymeric chains. SUMO chains are detectable under normal, unstressed conditions and increase markedly in response to RS. These chains can target proteins for proteasomal degradation by recruiting SUMO-targeted ubiquitin ligases (STUbL)[20,21].

Each eukaryotic genome contains difficult-to-replicate regions, including centromeres—essential chromatin loci that facilitate the assembly of the multiprotein kinetochore complex and are thus crucial for accurate DNA segregation. In fission yeast, centromeres are repetitive structures, resembling human centromeres. A central non-repetitive core region (cnt) is flanked by inverted innermost repeats (imr) and a series of outer repeats (otr)[22]. The H3 histone variant CENP-A (a kinetochore-specific histone), called Cnp1 in Schizosaccharomyces pombe, is localized at cnt and, to some extent, imr, but is excluded from otr[23]. The outer repetitive centromeric region is constitutively silenced by heterochromatin[24]. Centromeres are closely linked to SUMOylation; in particular, polySUMOylation of Constitutive Centromere Associated Network proteins leads to their delocalization from centromeres[25,26]. Interestingly, human RAD51 was shown to promote centromere stability by regulating CENP-A loading into the core region[27]. In S. pombe, loss of SUMOylation has detrimental effects on heterochromatin formation at the centromeric locus[28].

The RSR was proposed to rely on SUMOylation waves that target multiple components of replisomes (SUMO clouds)[19]. Here, we investigated the role of SUMO chains in the DNA metabolism of fission yeast. We found that the loss of SUMO chains leads to elevated, endogenous RS and spontaneous DNA damage, accompanied by increased recombination events. Hotspots of replication arrest and Rad51/Rad52 enrichment in SUMO chain-deficient cells occur at centromeres. We optimized SUMO-ID[29] proteomics in fission yeast to identify potential components of SUMOylation clouds at the sites of Rad52-dependent repair. This approach enabled us to identify PCNA as a polySUMOylation target that limits recombination at centromeres by modulating the activation of PRR pathway. Furthermore, we show that artificial attachment of SUMO chains to Rad52 abolish centromeric recombination. Our work provides evidence that SUMO chains are essential for resolution of spontaneous RS at centromeres.

## Results

### SUMO chains diminish endogenous replication stress

To investigate the role of SUMO chains in DNA metabolism, we used a SUMO variant in which all lysines were substituted with arginines (pmt3-KallR$^{SUMO-KallR}$), allowing conjugation to substrates but preventing chain formation (Supplementary Fig. 1a). The main SUMO-chain acceptor sites (K14, K30) are located in the disordered N-terminal region of Pmt3[30]. SUMO-KallR mutation did not impair growth compared to the SUMO-devoid strain (pmt3Δ$^{SUMOΔ}$, Supplementary Fig. 1b). SUMO-KallR variant exhibited a reduction in high molecular weight (HMW) SUMO conjugates, as previously reported for SUMO-K14,30R[30,31], but showed efficient substrate conjugation without accumulation of free mono-SUMO compared to WT strain (Supplementary Fig. 1c, red arrow).

Elevated RS and spontaneous DNA damage lead to the accumulation of ssDNA and repair proteins at arrested RFs. Loss of SUMO chains led to elevated Ssb3-YFP (RPA) foci in untreated SUMO-KallR cells, though less than in SUMOΔ (Fig. 1a and Supplementary Fig. 1d). We also noticed accumulation of Rad52 foci in SUMO-KallR mutant, indicating increased spontaneous DNA damage or compromised disassembly of protein complexes at DNA damage foci (Fig. 1b and Supplementary Fig. 1e). We did not observe any significant changes in Ssb3, Rad52, or Rad51 protein levels by Western blotting (Supplementary Fig. 1f–h). Spontaneous DNA damage results in DNA damage checkpoint activation, and SUMO mutants showed autophosphorylation of Chk1 kinase (Fig. 1c). We also assessed Rad51-DNA binding and observed increased Rad51 foci in the SUMO-KallR mutant by immunofluorescence (Fig. 1d, e). To explore whether SUMO chain-deficiency directly enhances endogenous RS, we verified EdU incorporation into replicating DNA in WT and SUMO-KallR strains. EdU signal in non-septated cells marks delayed DNA replication into the G2 phase. Loss of SUMO chains significantly increased the number of non-S-phase cells undergoing DNA synthesis compared to WT (Fig. 1f, g). These results demonstrate that SUMO chains are important for mitigating endogenous RS and DNA damage.

### SUMO chains fine-tune recovery from exogenous replication stress

Loss of SUMO chains leads to sensitivity to RS-generating drugs (methyl methanesulfonate (MMS) and hydroxyurea (HU)) but not to phleomycin-induced DSBs or UV-mediated DNA damage (Supplementary Fig. 2a). To determine whether SUMO chains contribute to the resolution of replication intermediates, we performed pulsed-field gel electrophoresis of intact chromosomes from HU-treated cells. WT completed genome duplication during the course of the experiment, whereas the SUMO-KallR mutant displayed delayed resolution of replication intermediates (Supplementary Fig. 2b, c). The SUMOΔ strain was unable to resolve replication intermediates throughout the experiment (Supplementary Fig. 2b, c). Flow cytometry analysis of cell cycle from samples taken in parallel to PFGE revealed that SUMO-KallR mutant progressed more slowly through S phase compared to WT (Supplementary Fig. 2e, red arrows). SUMOΔ mutant showed poor synchronization in HU and a markedly disrupted transition to G2. Monitoring recovery from HU-arrest at 10-min intervals confirmed delayed S-phase progression in the SUMO-KallR mutant (Supplementary Fig. 2f). These findings indicate that SUMO chains are required for efficient recovery from drug-induced replication arrest.

Replication restart rely on HR factors and SUMO alleles are known to show synthetic lethality with recombination mutants[30]. Although SUMO-KallR was viable with rad51Δ or rad52Δ, the double mutants were more sensitive to RS than the single mutants

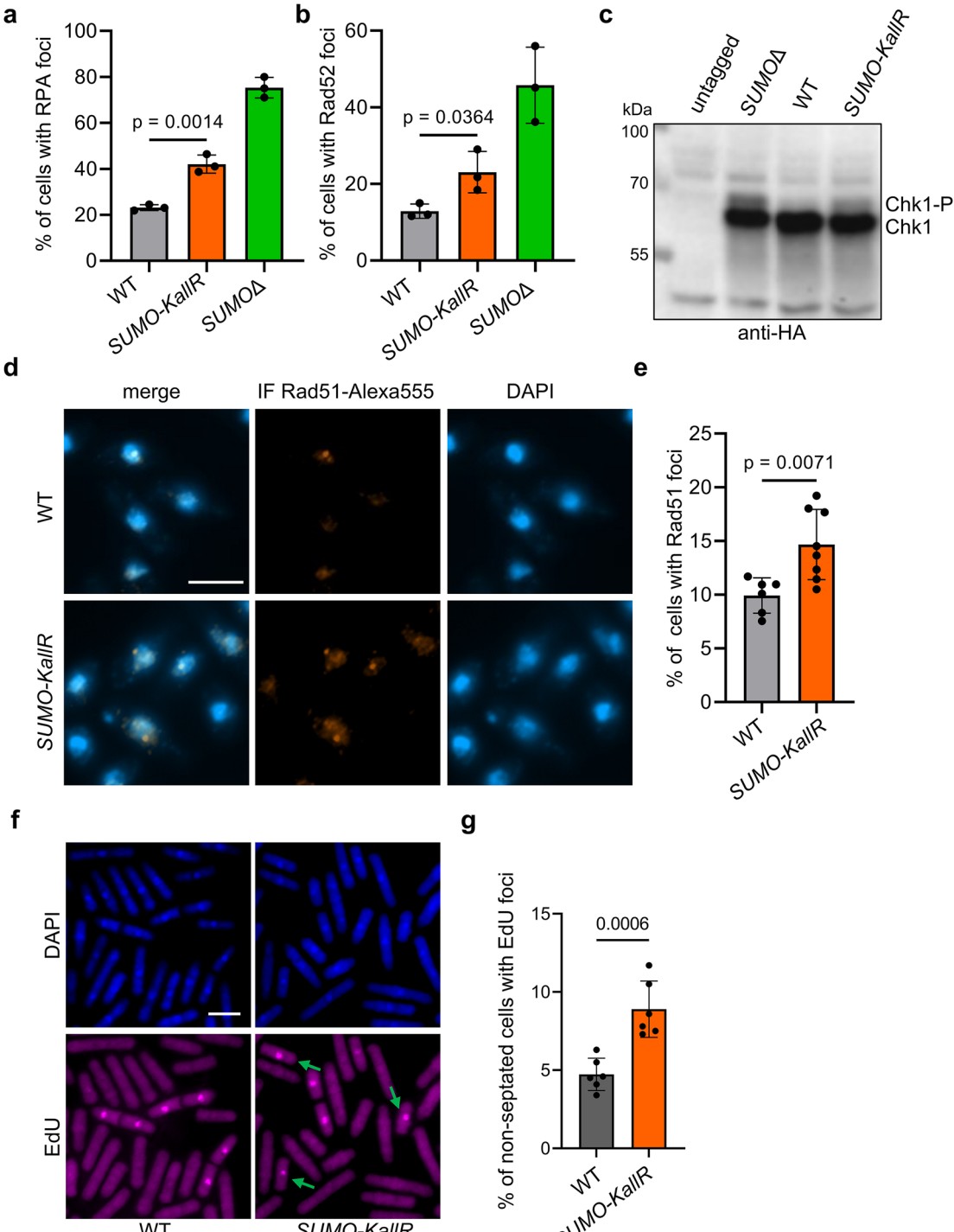

**Fig. 1 | Loss of SUMO chains leads to elevated replication stress and spontaneous DNA damage. a**, **b** % of cells forming RPA (Ssb3-YFP) and Rad52-YFP foci, respectively, in indicated strains. Dots represent values obtained from three independent biological experiments. Error bars show the standard deviation (SD) about the mean values. At least 500 cells were analyzed for each strain per single experiment. Two-sided Student's *t*-test was used to calculate the *p*-values. **c** The cell cycle DNA damage checkpoint activation in indicated strains. Autophosphorylation of Chk1-HA kinase (Chk1-P, phosphorylated form visible as slower migrating band) was confirmed by Western blot with anti-HA antibody of total protein extracts isolated from exponentially growing cultures. An untagged WT strain was included as a control for antibody specificity. **d** Example of immuno-fluorescence images of logarithmic WT and *SUMO-KallR* cells expressing native Rad51 probed with anti-Rad51 (primary) and anti-Rabbit Alexa Fluor 555 (secondary) antibody. Chromatin was stained with DAPI fluorescent dye. Scale bar = 5 μm.

**e** quantification of cells forming Rad51 foci in indicated strains (shown as %). Dots represent values obtained from independent biological experiments (*n* = 6 for WT and *n* = 8 for *SUMO-KallR*). At least 600 cells were analyzed for each strain for a single experiment. Error bars show the SD about the mean values. Two-sided Student's *t*-test was used to calculate the *p*-values. **f** Example of immunofluorescence images of EdU incorporation in logarithmic, septated (S-phase) and non-septated (G2 phase), WT and *SUMO-KallR* cells. The position of nuclei was determined by DAPI staining. Scale bar = 10 μm. **g** quantification of non-septated cells forming EdU foci in G2 phase, marking delayed DNA replication, occurring outside S-phase in indicated strains (shown as %). Dots represent values obtained from 6 independent biological experiments. At least 600 cells were analyzed for each strain for a single experiment. Error bars show the SD about the mean values. Two-sided Student's *t*-test was used to calculate the *p*-values. Source data are provided as a Source data file.

(Supplementary Fig. 3a, b). Considering enhanced endogenous DNA damage in *SUMO-KallR* and its genetic interaction with HR genes, we examined Rad52 foci dynamics after MMS treatment. While *SUMO-KallR* cells showed more Rad52 foci in logarithmic culture, MMS exposure equalized foci numbers between WT and mutant (Supplementary Fig. 3c). In WT, the number of Rad52 foci increased further after 2 h of growth in fresh, drug-free medium, consistent with ongoing repair of DNA damaged by MMS. Upon recovery, WT cells gradually resolved Rad52 foci, whereas *SUMO-KallR* cells induced more persistent Rad52 foci after drug-removal (Supplementary Fig. 3c). Chk1 phosphorylation upon MMS treatment was not altered in the absence of SUMO chains (Supplementary Fig. 3d). During the course of microscopy experiment, the Chk1-P was not shut down, with no major differences between WT and *SUMO-KallR* mutant (Supplementary Fig. 3e). These results suggest that SUMO chains facilitate recovery from RS and resolution of recombination repair centers.

### SUMO chains promote correct organization of centromeres by restricting Cnp1$^{CENP-A}$ deposition to central domains of centromeres

Given the increased spontaneous RS in *SUMO-KallR* mutant, we examined RPA (Ssb3-YFP) colocalization with centromeres, a well-known difficult-to-replicate locus. Using Mis6-RFP (a CENP-I ortholog, marking the inner kinetochore[32]) in WT and *SUMO-KallR* backgrounds (Fig. 2a), we observed a twofold increase in RPA-centromere colocalization in *SUMO-KallR* mutant (~25% of cells) compared to WT (~12% of cells, Fig. 2b). This raised the possibility that centromeres in the *SUMO-KallR* mutant may suffer from local RS. First, we examined the resistance of strains to thiabendazole (TBZ), a microtubule poison that decreases viability of strains with disrupted structure/maintenance of centromeres[33]. While the *SUMOΔ* strain was inviable on TBZ, *SUMO-KallR* mutant showed increased TBZ sensitivity compared to WT (Fig. 2c). Since SUMO is implicated in heterochromatin assembly at centromeres[28], we examined whether SUMO chains influence this process. Using *ura4+* or *ade6+* reporters inserted into centromeric *dg* repeats of the *otr1* (Supplementary Fig. 4a)[34], we showed that introduction of *SUMO-KallR* into the *swi6Δ* background (abolished heterochromatin formation[35]) or WT did not affect the growth on 5-FOA and selective media (Supplementary Fig. 4b, c). As expected, *swi6Δ* and *swi6Δ SUMO-KallR* expressed *ura4+* gene, in contrast to WT and *SUMO-KallR* alone (Supplementary Fig. 4d), indicating that heterochromatin formation was unaffected by the loss of SUMO chains.

Improper loading of CENP-A core histone disrupts centromeric organization and leads to chromosomal instability[36]. To verify whether centromeric structure is altered in the *SUMO-KallR* mutant, we used fluorescent microscopy to examine centromeric foci of CFP-Cnp1$^{CENP-A}$ in logarithmic *S. pombe* cells. In WT cells, single dots corresponding to clustered centromeres at the spindle pole body (SPB) were observed, while in *SUMO-KallR*, aberrant, elongated CFP-Cnp1 foci appeared more frequently (Fig. 2d, e). This suggested that Cnp1$^{CENP-A}$ may spread to ectopic sites. This alteration in Cnp1 distribution was not associated with changes in protein levels (Fig. 2f). To directly examine Cnp1 distribution at centromeres, we performed chromatin immunoprecipitation (ChIP) of CFP-Cnp1 from WT and *SUMO-KallR* cells, and found significantly more Cnp1 in the *SUMO-KallR* mutant, not only in the *cnt* region, but also within the *dg* outer repeats (Supplementary Fig. 5a, b). To further characterize Cnp1 distribution and assess its global localization in the *SUMO-KallR* mutant, we used a calibrated ChIP-seq approach[37]. On all three chromosomes, *SUMO-KallR* cells showed excessive Cnp1 binding to the central centromeric domain and spreading into the entire *otr* region (Fig. 2g and Supplementary Fig. 5c, d, compare blue signal of *SUMO-KallR* vs red corresponding to WT; green line marks untagged control). No additional Cnp1 binding sites were detected elsewhere on the chromosomes (Supplementary

Fig. 6a–c). Thus, we concluded that the loss of SUMO chains led to the spreading of Cnp1$^{CENP-A}$ to outer repetitive sequences of centromere.

### Slx8 STUbL does not play a major role in Cnp1$^{CENP-A}$ regulation in *S. pombe*

Given the recent report, showing Slx8 colocalization with centromeric locus in *S. pombe*[38], we sought to determine whether Slx8 STUbL-mediated degradation of polySUMOylated proteins contributes to the observed disruption of centromere organization in *SUMO-KallR*. We used the *slx8-29* temperature-sensitive mutant. At the permissive temperature (25 °C), *slx8-29* did not accumulate HMW SUMO conjugates (Supplementary Fig. 7a). Upon shift to the restrictive temperature (34 °C), *slx8-29* cells accumulated HMW SUMO conjugates, showed reduced viability, and became hypersensitive to HU, MMS, and TBZ, in a manner dependent on SUMO chains (Supplementary Fig. 7a, b). In budding yeast, the Slx5-Slx8 STUbL pathway was implicated in regulating the loading and stability of Cse4 (the *S. cerevisiae* ortholog of Cnp1). To assess whether a similar mechanism exists in *S. pombe*, we performed a cycloheximide (CHX) chase experiment at 34 °C. We did not observe a marked decrease in Cnp1 levels in either WT or analysed mutants. *SUMO-KallR* clearly showed a WT-like profile (Supplementary Fig. 7c). We next analysed CFP-Cnp1 localization using live-cell imaging (Supplementary Fig. 7d). At the permissive temperature (25 °C, blue), we observed increased aberrant Cnp1 foci exclusively in *SUMO-KallR* and *slx8-29 SUMO-KallR* strains. Upon shifting to 34 °C, the number of aberrant foci increased not only in *SUMO-KallR*, but also in *slx8-29*. Interestingly, the double mutant *slx8-29 SUMO-KallR* showed Cnp1 aberrant levels comparable to those in each single mutant (Supplementary Fig. 7e). These data suggest that both, SUMO chains attachment and removal of polySUMOylated proteins by STUbL may be involved in proper CENP-A localization.

### Recombination-mediated maintenance of centromeres depends on formation of SUMO chains

The increased RPA accumulation at centromeres in the absence of SUMO chains (Fig. 2a, b) suggested enhanced recruitment of recombination factors. ChIP analysis confirmed elevated binding of Rad52-YFP and Rad51 to centromere cores in the *SUMO-KallR* mutant (Fig. 3a, b). Since MMS-induced Rad52 foci often colocalize with Sad1, a SPB protein anchoring centromere[39], we examined spontaneous Rad52-Sad1 colocalization. It was significantly more frequent in the *SUMO-KallR* mutant, supporting increased centromeric engagement of repair factors (Supplementary Fig. 8).

Loss of SUMO chains also enhanced TBZ sensitivity in *SUMO-KallR rad51Δ* and *SUMO-KallR rad52Δ* double mutants compared to single mutants (Fig. 3c). It indicates that HR and SUMO chains might contribute to centromere stability at least partially through independent pathways. Interestingly, *SUMO-KallR* allele was epistatic to the *rad51-3A* separation-of-function mutant (defective in strand exchange, but proficient in DNA binding[6]), suggesting potential involvement of polySUMOylation in regulation of Rad51 catalytic activity in the context of centromere maintenance (Fig. 3c). We hypothesized that excessive binding of recombination factors to centromeres results from potential site-specific RFs stalling or DNA damage that could lead to increased centromeric recombination. To test it, we used the *ade6B/ade6X* system[40] (Fig. 3d). *SUMO-KallR* cells showed a significantly elevated recombination frequency at centromeres (Fig. 3e). We also measured the rate of mitotic recombination on chromosome arm using the *ade6-L469-his3-ade6-M375* assay (Fig. 3f)[41]. Strikingly, we found a decreased rate of intrachromosomal recombination in *SUMO-KallR* compared to WT cells (Fig. 3g). These findings indicate that SUMO chains limit recombination specifically at centromeres.

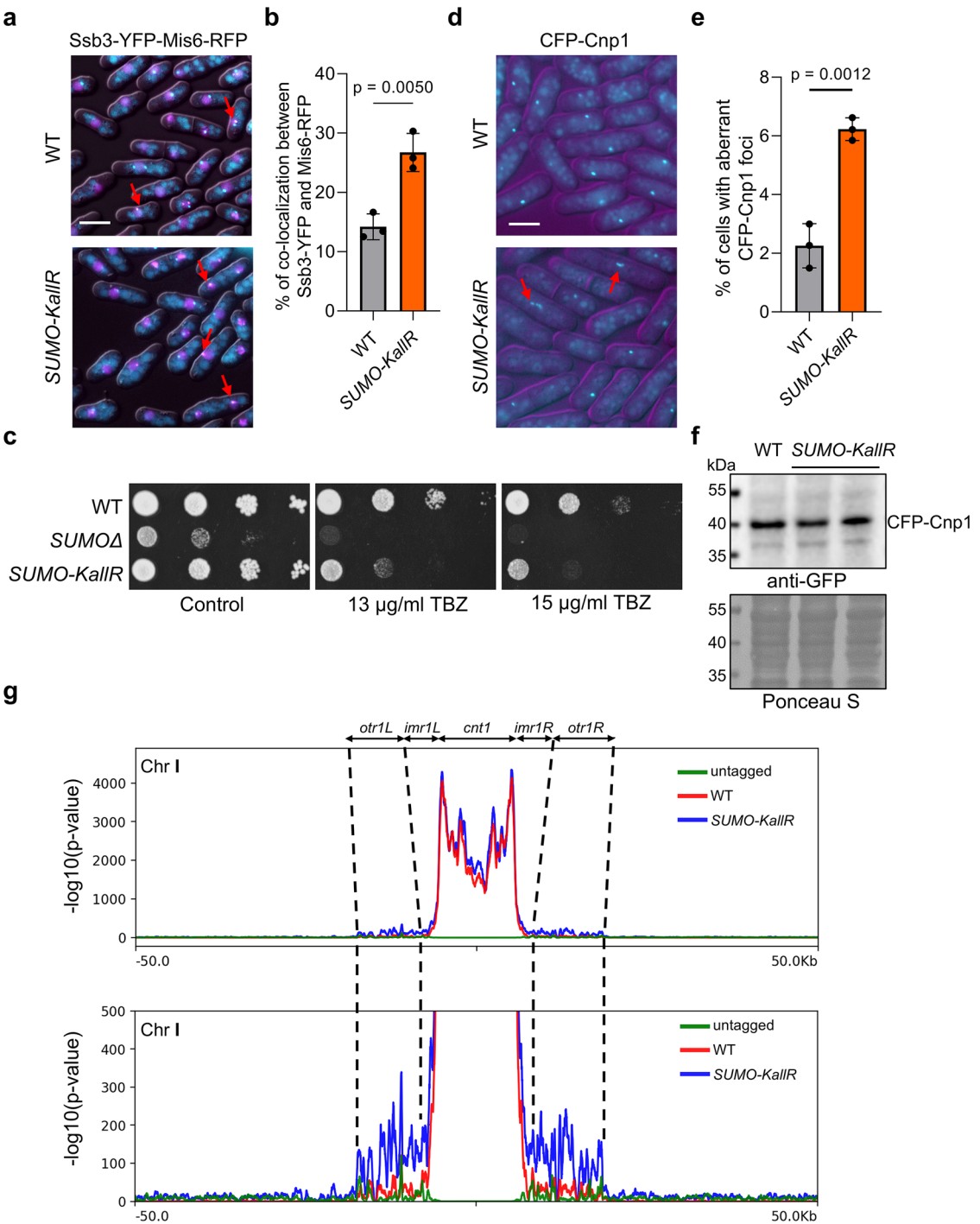

**Fig. 2 | SUMO chains control centromere maintenance. a** Examples of co-localization between RPA and centromere as a merged bright-field (DIC), with fluorescence images (RFP, YFP) of logarithmically growing WT and *SUMO-KallR* cells expressing endogenous Mis6-RFP and Ssb3-YFP proteins. Red arrows indicate co-localization events between two proteins. Scale bar = 5 μm. **b** % quantification of co-localization between RPA and centromeres in cells of indicated strains. Dots represent values obtained from 3 independent biological experiments. In all, 200–300 cells were analyzed for each strain. Error bars show the SD about the mean values. Two-sided Student's *t*-test was used to calculate the *p*-values. **c** Cell growth assay of indicated strains. Tenfold serial dilution of exponential cultures were dropped on YES agar plates containing TBZ thiabendazole. **d** Example of fluorescence images of logarithmically growing cells expressing endogenous CFP-Cnp1 fusion protein in indicated strains. Red arrows indicate aberrant CFP-Cnp1 foci. Scale bar = 5 μm **e** % quantification of cells forming aberrant Cnp1 foci in indicated strains. Dots represent values obtained from 3 independent biological

experiments. In all, at least 500 cells were analyzed for each strain. Error bars show the SD about the mean values. Two-sided Student's *t*-test was used to calculate the *p* values. **f** Expression of CFP-Cnp1 analyzed by Western blot with anti-GFP antibody of total protein extracts isolated from exponentially growing cultures of indicated strains. A Ponceau S-stained blot shows equal amounts of total proteins loaded onto the gel. Representative blot from *n* = 3 biological replicates. **g** Calibrated ChIP-seq profiles of CFP-Cnp1 around fission yeast centromere on chromosome I in indicated strains. Logarithmic cultures of indicated strains were mixed with budding yeast calibrator aliquot (exponential culture of Scc1-Pk) and extracted DNA was used to construct DNA libraries and NGS analysis. The central part of *cnt* was set as 0. The range of central domain and *otr* region are designated with black dashed lines. The upper panel presents locus around the centromere with values on *Y*-axis, that present -log10(*p*-value) set up to 4000 to cover the signal from the core of centromere, whereas bottom panel scale on *Y*-axis was set to 500 to clearly examine the enrichments of Cnp1. Source data are provided as a Source data file.

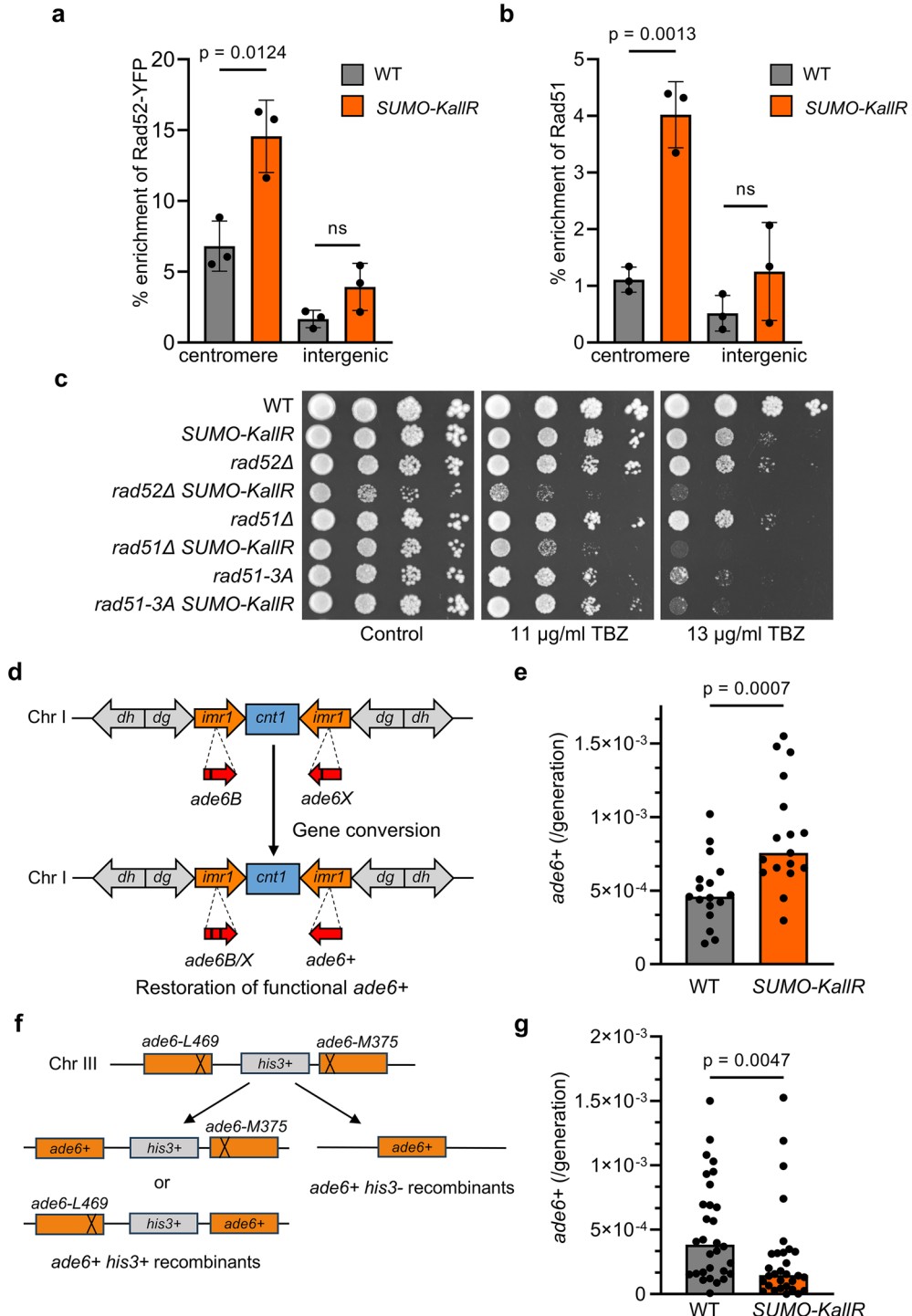

**Fig. 3 | SUMO chains limit excessive recombination at centromeres. a** Binding of Rad52-YFP to the centromere locus in indicated strains analyzed by ChIP-qPCR using anti-GFP antibody (relevant primers in Table S2). Intergenic locus on chromosome II was used as an internal specificity control. Dots represent values obtained from 3 independent biological repeats. Error bars: SD about the mean values. Two-sided Student's *t*-test was used to calculate the *p*-values. ns non significant. **b** Rad51-binding to the centromere locus in indicated strains analyzed by ChIP-qPCR using anti-Rad51 antibody, Normal Rabbit IgG antibody, and relevant primers (Table S2). Intergenic locus was used as an internal specificity control. Dots represent values obtained from 3 independent biological repeats. Error bars: SD about the mean values. Two-sided Student's *t*-test was used to calculate the *p*-values. **c** Drop dilution assay of indicated strains. Tenfold serial dilution of exponential cultures were dropped on YES agar plates containing TBZ in indicated concentrations. **d** A scheme of centromeric gene conversion between *ade6B* and

*ade6X* heteroalleles introduced into *imr1* inverted repeats at *cen1*. Restoration of functional *ade6+* gene was measured on selective plates. **e** Spontaneous rate of centromeric gene conversion described in (**d**) in saturated cell cultures of indicated strains. Dots represent values obtained from 17 independent cultures from single colonies. Median is marked for each strain. Two-sided Mann−Whitney *U* test was used to calculate the *p*-values. **f** A scheme of intrachromosomal mitotic recombination between *ade6-L469* and *ade6-M375* heteroalleles separated by *his3+* gene introduced into native *ade6* locus on chromosome III. Recombination between inverted *ade6* mutated alleles can produce deletion-type and conversion-type *ade6+* recombinants. **g** Spontaneous rate of intrachromosomal mitotic recombination described in (**f**) in saturated cell cultures of indicated strains. Dots represent values obtained from 30 independent cultures from single colonies with marked median. Two-sided Mann−Whitney *U* test was used to calculate the *p*-values. Source data are provided as a Source data file.

## Split-SUMO-ID proteomics reveals SUMO-dependent Rad52 interactors

Given the significant increase of Rad52 foci in unchallenged *SUMO-KallR* cells, and its enrichment at centromeres, we adapted the SUMO-ID proteomics approach, originally developed for human cells[29] to identify the content of potential SUMO clouds arising at the Rad52 repair sites. We used a split-TurboID biotin ligase system, dividing it into NTurbo (1–78) and CTurbo (79–320), which reconstitute functional activity upon proximity[42]. NTurbo was fused with a 6HA-tag to N-terminus of Pmt3$^{SUMO}$ or Pmt3-KallR$^{SUMO-KallR}$ under the *nmt41* promoter and cloned into the pREP41 plasmid. CTurbo, along with a 12Pk tag, was fused to C-terminus of endogenous Rad52 (Fig. 4a). Overexpression of the NTurbo-SUMO or NTurbo-SUMO-KallR was non-toxic and partially suppressed *SUMOΔ* sensitivity to MMS (Supplementary Fig. 9a). Rad52-CTurbo strain exhibited no apparent sensitivity to genotoxic agents and mirrored the *SUMO-KallR* phenotypes when introduced to this background, in contrast to highly sensitive *rad52Δ* mutant (Supplementary Fig. 9b, c). Following 21 h of growth in biotin-free media, cells were treated with 50 μM biotin for 3 h. Biotinylation was markedly increased only in the strains co-expressing both TurboID subunits, confirming reconstitution of biotin ligase (Supplementary Figs. 9d and 10a). Western blotting verified expression of the fusion proteins (Supplementary Figs. 9e and 10b) and streptavidin pulldown confirmed specific enrichment of biotinylated proteins in the samples co-expressing both subunits of TurboID (Fig. 4b and Supplementary Fig. 10c). The membrane was then stripped and developed against Pk or HA antibodies to show split-TurboID components (Fig. 4b and Supplementary Fig. 10c). While some unspecific binding of 6HA-NTurbo-SUMO or Rad52-12Pk-CTurbo to beads may reflect endogenous Bpl1 biotin ligase activity, encoded by an essential gene, the strongest biotinylation was observed only with both NTurbo and CTurbo expressed. Streptavidin-bound proteins from samples SUMO-ID or SUMO-KallR-ID (expressing Rad52-CTurbo and NTurbo-SUMO or NTurbo-SUMO-KallR) were subjected to a bottom-up mass spectrometry analysis in conjunction with samples coming from appropriate control experiments (expression of sole TurboID subunits in fusion with bait proteins: NTurbo-SUMO, NTurbo-SUMO-KallR or Rad52-CTurbo). Reproducibility analyses (Pearson correlation of replicates and principal component analysis) are presented on Supplementary Fig. 11a, b. Both SUMO and Rad52 proteins were enriched in SUMO-ID as well as SUMO-KallR-ID samples, validating successful purification of the SUMO/Rad52-dependent interactome. The WT SUMO-ID dataset (volcano plot, Fig. 4c) showed significant enrichment of RFs components, including polymerase delta subunit Pol3 and Pcn1 (PCNA). Among enriched proteins, we found chaperone/chromatin remodeler Rvb2, already reported as an interactor of Rad52[43]. We also detected histone H2B (Htb1) and checkpoint proteins Rad24 and Rad25, that not only modulate the DNA repair, but were also shown to interact physically with the proteasomes[44]. Importantly, some of the identified hits, including Pol3, Rvb2, or Htb1 are known SUMO targets[45]. Despite the increased biotinylation signal in samples from *SUMO-KallR* background after streptavidin pulldown (Supplementary Fig. 10c), which suggests a potential impact of polySUMOylation on restricting the activity of endogenous biotin ligase, mass spectrometry revealed a marked reduction in the Rad52- and SUMO-dependent interactome. Particular hits enriched in the WT background (Pcn1, Pol3, Rvb2, Htb1, Rad24, and Rad25) were not significantly enriched in *SUMO-KallR* mutant (Fig. 4d). Overall, we concluded that Rad52-dependent repair might take place at the sites of replication arrest and proteins involved in the restart of DNA synthesis could undergo polySUMOylation which promote transient interactions during the repair process.

## PCNA—a SUMOylation target responsible for excessive recombination in *SUMO-KallR* mutant

PCNA SUMOylation has been extensively studied in budding yeast and human cells[45,46]. However, it remained unclear whether the fission yeast ortholog could also be modified by SUMO. To assess PCNA SUMOylation in *S. pombe*, we performed Ni-NTA pulldown of overexpressed Pcn1-6His-3Flag (Pcn1-HF) in WT, *SUMO-KallR*, and E3 SUMO ligase mutants (*pli1Δ* and *nse2RINGΔ*, viable mutant with deletion of 56 C-terminal amino acids, including C195 and H197 conserved residues, thus inactive as a SUMO ligase). We successfully precipitated Pcn1-HF and showed that it can undergo SUMO modification in WT (Fig. 5a). Interestingly, while mono-SUMOylation of Pcn1 was consistently observed, the appearance of higher-molecular-weight bands depended on Pli1, not Nse2. These higher SUMO-conjugated forms were absent in extracts from *SUMO-KallR* and *pli1Δ* mutants, suggesting attachment of SUMO chains to PCNA.

Given that PCNA-K164 is a known SUMO/ubiquitin site in budding yeast and humans[8], and is essential for ubiquitination in fission yeast[47], we tested SUMOylation and ubiquitination of Pcn1-HF and Pcn1-K164R-HF by Ni-NTA pulldown. Pcn1-HF underwent both SUMOylation and ubiquitination. In contrast, Pcn1-K164R-HF showed a markedly reduced ubiquitin signal, but slightly increased SUMOylation compared to Pcn1-HF (Fig. 5b).

To test whether PRR contributes to *SUMO-KallR* phenotypes, we generated *pcn1-K164R SUMO-KallR* double mutants. *SUMO-KallR* allele was increasing sensitivity of *pcn1-K164R* upon global RS induced by MMS. However, under TBZ exposure, *pcn1-K164R* mutation was beneficial for cells devoid of SUMO chains (Fig. 6a). It raised the possibility that unbalanced PRR activation may cause TBZ sensitivity in *SUMO-KallR* mutants. Supporting this, deletion of *rad8* + E3 ubiquitin ligase, which is required for TS activation, suppressed *SUMO-KallR* TBZ sensitivity similarly to *pcn1-K164R*, but increased sensitivity of *rad8Δ* to MMS. In contrast, deletion of TLS polymerases, Rev1 or Rev3, did not suppress *SUMO-KallR* TBZ sensitivity (Fig. 6a). We then examined PCNA modification by Ni-NTA pulldown in WT and *SUMO-KallR* strains and found that loss of SUMO chains in *SUMO-KallR* led to increased polyubiquitination of Pcn1-HF (Fig. 6b and Supplementary Fig. 12a). This hyper-ubiquitination depended on Rad8 activity, as shown by analysis of Pcn1-HF in *rad8Δ SUMO-KallR* (Fig. 6c and Supplementary Fig. 12b). Since suppression of *SUMO-KallR* by *rad8Δ* or *pcn1-K164R* was only observed under TBZ, we assessed centromeric recombination rates. The *pcn1-K164R* mutation reduced the elevated recombination seen in *SUMO-KallR* cells (Fig. 6d), suggesting that SUMO chains on PCNA contribute to proper activation of TS, which may be the preferred repair mechanism at centromeres. To support this, we monitored Rad52-GFP colocalization with the centromere marker Mis6-RFP and found that enhanced Rad52-centromeres interaction in *SUMO-KallR* was reduced in the *pcn1-K164R SUMO-KallR* double mutant (Fig. 6e and Supplementary Fig. 12c). However, despite reduced centromeric recombination, the aberrant Cnp1 morphology persisted in the *SUMO-KallR pcn1-K164R* strain (Fig. 6f and Supplementary Fig. 12d).

## PolySUMOylation of Rad52 safeguards centromeric stability

spRad52 was reported as a SUMOylation target[48]. Bioinformatics analysis of Rad52 sequence using GPS SUMO 2.0[49] tool showed a high probability of its SUMOylation, particularly at the core and C-terminus (Supplementary Fig. 13a). To test the impact of SUMO chains on Rad52 we engineered a Rad52 variant fused at its C-terminus to four active Pmt3-GG particles and a 6his-3Flag (referred as Rad52-S$_{chain}$, Fig. 7a), similarly to artificial SUMO targets in budding yeast, where 4 SUMO units organised as a linear chain were sufficient to attract the STUbL

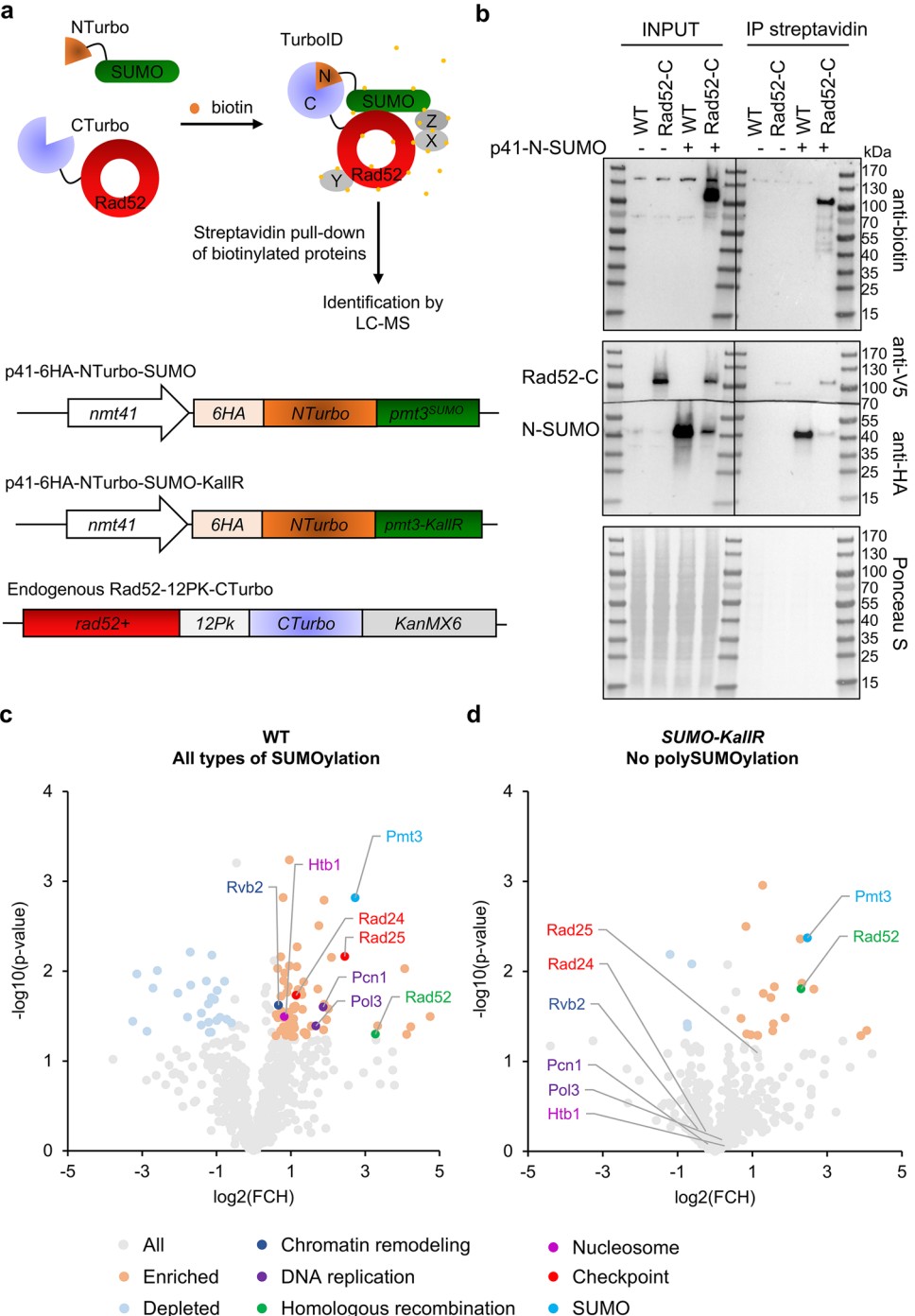

**Fig. 4 | Analysis of Rad52- and SUMO-dependent interactome. a** A schematic drawing depicting identification of potentially SUMOylated proteins at the sites of Rad52 repair by Split-TurboID (upper panel). In this strategy, the N-terminal, small Split-TurboID fragment is fused to SUMO or SUMO-KallR, and the C-terminal, large fragment to a protein of interest (here Rad52). Target SUMOylation (or SUMO-SIM interaction) results in protein-fragment complementation and reconstitution of the TurboID enzyme, allowing specific biotin proximity labeling of interactors, and other proximal proteins, in a SUMO-dependent manner. Biotinylated proteins (X, Y, Z) can then be enriched by streptavidin purification and identified by mass spectrometry (MS). In the lower panel, schematic representation of the genetic structure of Split-TurboID tagged constructs is also shown. **b** Streptavidin pulldown of biotinylated proteins in indicated transformants. Western blot analysis of total cell extracts (INPUT, left side) and streptavidin pulldown (IP, right side) prepared from a control untagged (WT) and the Rad52-12Pk-CTurbo (Rad52-C) strains transformed with an empty vector or 6HA-N-Turbo-NSUMO (N-SUMO) bearing plasmid was performed using anti-biotin (upper panel), and then the membrane was

stripped, cut and developed with anti-V5 (middle panel), and anti-HA (lower panel) antibody. A Ponceau S-stained blots were added to show equal amounts of total proteins loaded onto the gel after isolation from individual transformant. Representative blots from $n = 3$ biological replicates. All INPUTs and IPs blots are cut and merged from shorter and longer exposures of the same membrane. **c, d** Results of three independent biological experiments followed by proteomics analyses of the proximal biotinylation assay obtained for NTurbo-SUMO-ID (**c**) or NTurbo-SUMO-KallR-ID (**d**) in the WT and *SUMO-KallR* background, respectively. Proteins significantly enriched in comparison to controls (top right-hand side) are identified as possible Rad52 interactors and potential SUMOylation targets at endogenous replication stress sites. All hits were annotated according to their cellular function, process, and location based on GO Terms annotations database. Annotations were then slimmed into broader categories, and the categories of interest are shown on the plot. Statistical significance was calculated using two-sided Student's *t*-test without an adjustment for multiple comparisons, but with an effect size cut-off applied. Source data are provided as a Source data file.

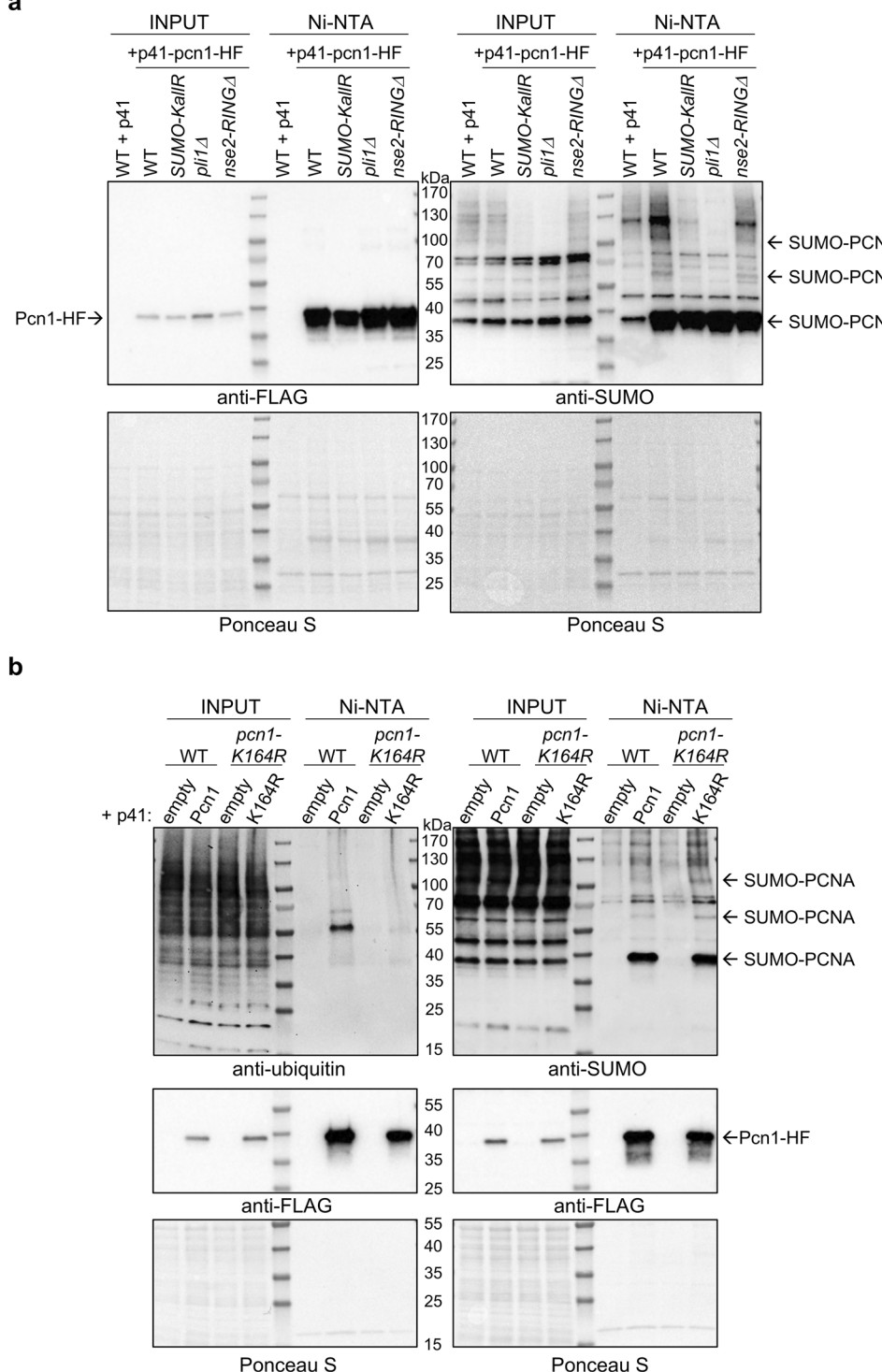

**Fig. 5 | *S. pombe* PCNA undergoes simultaneous SUMOylation and ubiquitination. a** SUMOylation status of Pcn1 from indicated strains transformed with pREP41-pcn1-6his-3Flag (p41-pcn1-HF) alongside with WT control transformed with empty pREP41-3Flag (p41). Cells were disrupted in denaturing conditions and subjected to Ni-NTA pulldown. Same samples were run on separate gels and Pcn1 precipitation was confirmed by anti-Flag Western blot (left panel). SUMOylation status of precipitated Pcn1 was examined with anti-SUMO antibody (right panel). Ponceau S was included for both Western blots to show the amount of protein loaded onto the gels. Representative blots from *n* = 3 biological replicates. **b** Ubiquitination and SUMOylation status of Pcn1-HF and Pcn1-K164R-HF from indicated strains. WT cells were transformed either with empty pREP41-3Flag vector (empty) or pREP41-pcn1-HF (Pcn1); *pcn1-K164R* mutant was transformed with empty pREP41-3Flag (empty) or pREP41-pcn1-K164R-HF (K164R). Cells from indicated transformants were disrupted in denaturing conditions and Ni-NTA pulldown was performed. Same samples were run on independent gels and membranes were developed against ubiquitin (left side), SUMO (right side), then stripped and developed against anti-Flag (middle panel). Ponceau S-stained membranes are shown for both panels to present the amount of protein loaded onto gels. Representative blots from *n* = 2 biological replicates. Source data are provided as a Source data file.

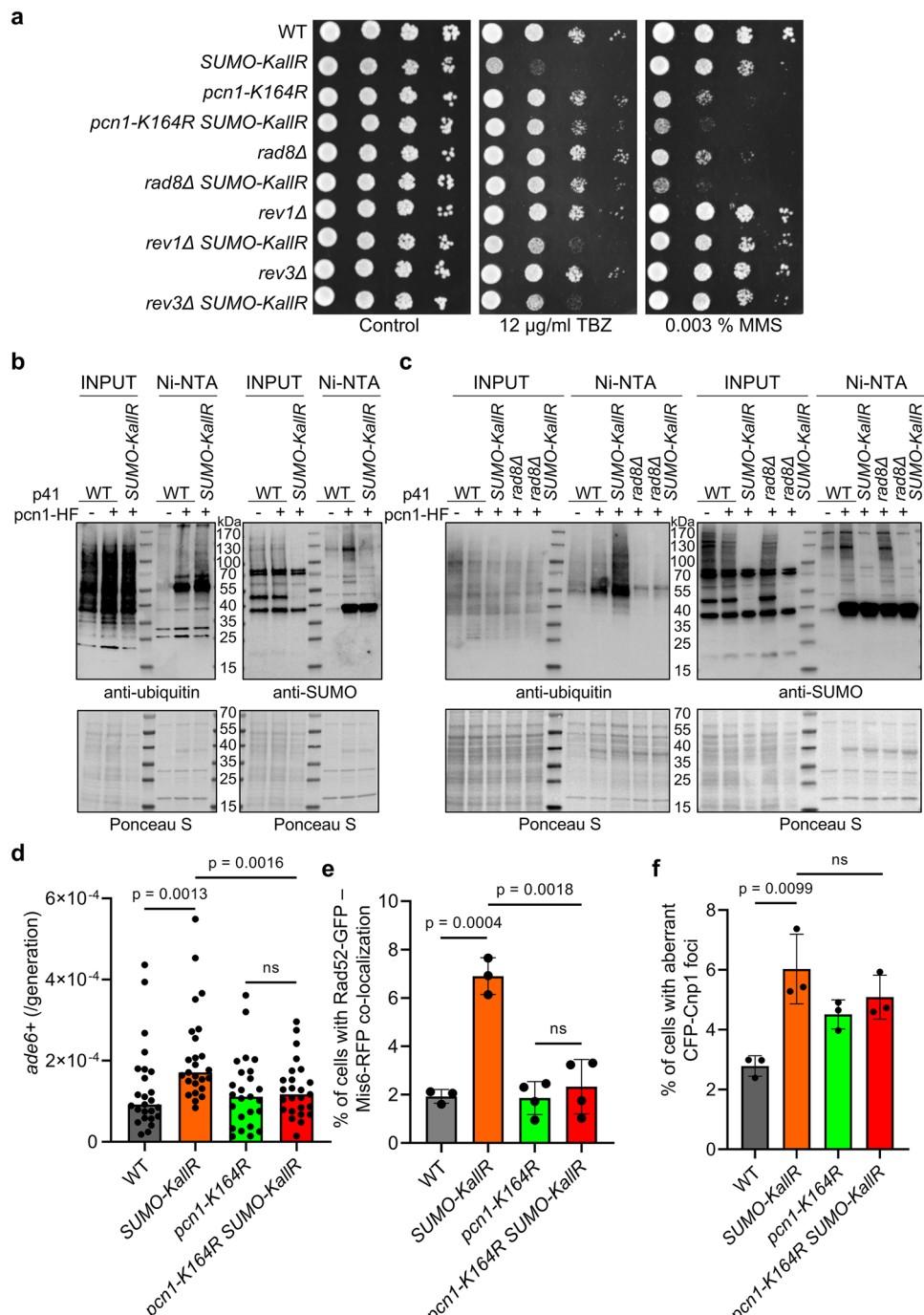

**Fig. 6 | Rad8-dependent polyubiquitination of PCNA is increased upon SUMO chains loss. a** Drop dilution assay of indicated strains. Tenfold serial dilution of exponential cultures were dropped on YES agar plates containing TBZ and MMS in indicated concentrations. **b** Ubiquitination and SUMOylation status of Pcn1-HF Ni-NTA-pulldown from WT and *SUMO-KallR* strains. WT transformed with empty pREP41-3Flag included as specificity control (marked as "−"), otherwise WT and *SUMO-KallR* were transformed with pREP41-pcn1-HF (p41-pcn1-HF, marked as "+"). Same samples were run to obtain two membranes, left panel was probed against ubiquitin, right panel was probed with anti-SUMO. Ponceau S are presented to show protein loading. Representative blots from *n* = 3 biological replicates. Anti-Flag blot, done on stripped membrane initially developed with anti-ubiquitin was performed to show amounts of precipitated Pcn1-HF is presented in Supplementary Fig. S12a. **c** Ubiquitination and SUMOylation status of Pcn1-HF Ni-NTA-pulldown of Pcn1-HF from indicated strains. WT transformed with empty pREP41-3Flag included as specificity control ("−"). "+" marks cells transformed with pREP41-pcn1-HF (p41-pcn1-HF). Same samples were run on independent gels, left panel was probed

against ubiquitin, right panel was probed with anti-SUMO. Ponceau S presented to show protein loading. Representative blots from *n* = 3 biological replicates. Anti-Flag blot was done after stripping of membrane developed initially with anti-ubiquitin to show amounts of precipitated Pcn1-HF and is presented in Supplementary Fig. S12b. **d** Spontaneous rate of centromeric gene conversion as in Fig. 3d, e done on saturated cell cultures of indicated strains. Dots represent values obtained from 25 independent cultures from single colonies. Median is marked for each strain. Two-sided Mann–Whitney *U* test was used to calculate the *p* values. **e** % of cells exhibiting co-localization between Rad52-GFP and Mis6-RFP. 3 biological experiments are presented, each time at least 500 cells were counted for each strain. Error bars show the SD about the mean values. Two-sided Student's *t*-test was used to calculate the *p*-values. **f** % of cells forming aberrant CFP-Cnp1 foci in indicated strains. Dots represent values obtained from 3 independent biological experiments. In all, at least 500 cells were analyzed for each strain. Error bars show the SD about the mean values. Two-sided Student's *t*-test was used to calculate the *p*-values. Source data are provided as a Source data file.

activity[50]. We obtained a stable genomic construct, expressing solely Rad52 with a linear SUMO chain, although its expression was lower compared to WT Rad52 (Supplementary Fig. 13b). Ni-NTA pulldown showed SUMOylation of control Rad52-6His-3Flag (referred to as Rad52-HF) and confirmed expression of Rad52-S$_{chain}$, with a ~100 kDa Flag band aligning with anti-SUMO signal, consistent with 4 SUMO units attached to Rad52 (Fig. 7b). Importantly, both WT Rad52-HF and Rad52-S$_{chain}$ were accumulating multiple, heavy bands corresponding to their extended polySUMOylation, that were largely diminished in the SUMO-KallR background. Rad52 polySUMOylation was dependent on activity of Pli1 E3 SUMO ligase (Fig. 7c). The rad52-S$_{chain}$ mutant exhibited milder sensitivity to MMS than rad52Δ mutant. Introduction of rad52-S$_{chain}$ to SUMO-KallR mutant partially suppressed its TBZ sensitivity (Fig. 7d). Strikingly, Rad52-S$_{chain}$ abolished centromeric recombination in both WT and SUMO-KallR backgrounds (Fig. 7e). Notably, immunofluorescence on rad52-S$_{chain}$ showed that it can reduce formation of Rad52 foci compared to WT and suppress excessive formation of repair foci in the absence of SUMO chains (Fig. 7f and Supplementary Fig. 13c). Given the observed suppression of SUMO-KallR phenotypes by pcn1-K164R or rad52-S$_{chain}$ alleles, we examined the genetic interaction between the pcn1-K164R and rad52+. First, we observed by tetrad dissection, that all the double mutants supposed to bear rad52Δ pcn1-K164R mutations were not able to germinate (Supplementary Fig. 14a). On the other hand, rad52-S$_{chain}$ markedly increased the sensitivity of pcn1-K164R mutant on MMS and TBZ (Supplementary Fig. 14b). We also found that the triple mutant pcn1-K164R rad52-S$_{chain}$ SUMO-KallR was exhibiting the same phenotype as the rad52-S$_{chain}$ pcn1-K164R strain on TBZ plates (Supplementary Fig. 14c). This suggests that PRR dependent on PCNA-K164 ubiquitination and Rad52 activity at centromeres, is independent of each other. Moreover, SUMO-KallR sensitivity to TBZ is no longer suppressed when both pcn1K164R and rad52-S$_{chain}$ are combined. However, similarly to pcn1-K164R mutation, Rad52-S$_{chain}$ did not rescue Cnp1 mislocalization in SUMO-KallR (Fig. 7g and Supplementary Fig. 14d). Thus, aberrant Cnp1 spreading in SUMO-KallR was not dependent on pcn1-K164 mutation or Rad52 polySUMOylation.

The inner kinetochore protein Cnp3 (ortholog of human CENP-C) was shown to stabilize Cnp1 and confine it to the core of centromere[51]. Consistently, cnp3+ mutations result in reduced Cnp1 levels at centromeres[51,52]. To further explore the relationship between centromere organization and SUMO chains, we combined deletion of cnp3+ with SUMO-KallR mutant. In unchallenged conditions, the double mutant SUMO-KallR cnp3Δ exhibited a significantly higher number of spontaneous Rad52-YFP foci, compared to either single mutant (Supplementary Fig. 15a, b). Interestingly, the double mutant SUMO-KallR cnp3Δ showed a reduction of spreaded Cnp1 foci (Supplementary Fig. 15c, d), while centromere declustering remained comparable to cnp3Δ single mutant (Supplementary Fig. 15e). We thus concluded that control of centromeric recombination and maintenance of centromere organization are independent of each other, yet both may undergo regulation by polySUMOylation.

## Discussion

SUMO chains are formed in eukaryotic organisms, but their precise effect on DNA metabolism remains unclear. Here, we reveal the beneficial impact of SUMO chains formation that diminish spontaneous RS and DNA damage. SUMO chains control recombination at difficult to replicate site−centromeres. We have found that SUMO chains attached to PCNA contribute to the accurate activation of PRR pathways at centromeres, which are employed without external DNA damage. Unbalanced channeling towards TS led to increased recruitment of Rad52 HR mediator into centromeric region, resulting in elevated recombination. Artificial SUMO chains attachment to Rad52 safeguards centromeres from excessive recombination resulting from SUMO-KallR mutation (see Fig. 8 for graphical summary).

Initial findings from budding yeast suggested that SUMO chains undergo dynamic removal by Ulp2 SENP protease[53]. Subsequent studies from budding yeast proved that SUMO chains-deficiency in smt3-allKR led to spontaneous DNA damage, RS, and distorted rDNA structure, visualized by fluorescent microscopy of rDNA markers[54,55]. A recent work by Wang group explored in details rDNA distortion upon SUMO chain loss and proved that polySUMOylation promotes the release of Tof2 from nucleolus for mitotic exit by extracting Tof2-SUMOchain for proteasomal degradation through Cdc48-STUbL pathway[56]. In fission yeast, it was shown that the SENP SUMO protease Ulp1 is a determinant for initiation of DNA synthesis at HR-restarted forks and that the proteasome contributes to sustaining the DNA synthesis at restarted RFs[57]. It extends a previous study, that showed SUMO chains as a block for HR-mediated DNA synthesis at site-specific replication arrest and underlined the need for re-localization of dysfunctional forks to nuclear pore complexes for Ulp1-mediated deSUMOylation[58]. Taken together, these results demonstrate that protein polySUMOylation may be the key regulatory signal in DNA metabolism. Here we show that, similarly to budding yeast, the loss of SUMO chains in pmt3-KallR mutant led to increase in spontaneous RS and DNA damage, as manifested by increase in RPA-bound ssDNA and Rad52 repair centers, particularly localized at centromeres. We hypothesized that Rad52 repair was dependent on local SUMO-mediated interactions. To explore the SUMO and Rad52-dependent interactome, we subjected Rad52 to SUMO-ID proteomics. In stress-free cultures of WT, we detected replication factors (Pol3, Pcn1) and histone H2B (Htb1), suggesting that Rad52-repair took place at the sites of replication arrest. Interestingly, when Rad52-SUMO-ID was performed in the pmt3-KallR$^{SUMO-KallR}$ background, it showed a decrease of identified proteins, which contrasts to results obtained in SILAC proteomics done for SMT3allKR, that showed overall accumulation of replication factors[59]. It was concluded that accumulation of replication factors in the absence of SUMO chains resulted from impaired degradation of replisome components. In our hands, Rad52-SUMO-ID performed in the pmt3-KallR$^{SUMO-KallR}$ background allowed detection of significantly enriched Pmt3-KallR and Rad52, but the majority of positive hits identified in WT SUMO-ID were not reaching statistical significance in the absence of SUMO chains. It would point that upon loss of SUMO chains, specific polySUMOylation events at Rad52 repair centers are largely diminished, and not all of them have to be related to proteasomal degradation. Otherwise, polySUMOylation constitutes a scaffold for transient interactions mediated by SIMs (SUMO-interacting motifs) and not necessarily all of detected proteins by the SUMO-ID are modified with SUMO, but some might be recruited to the sites of ongoing repair by SUMO-SIM mediated interactions that, which are promoted by formation of local SUMO clouds.

Detection of Pcn1 by Rad52-SUMO-ID draw our attention towards its potential SUMOylation. We directly examined the SUMOylation of overexpressed Pcn1-6His-3Flag by Ni-NTA pulldown and found that SUMO chains are counteracting the extent of PCNA ubiquitination. Thus, presence of SUMO chains on PCNA might contribute to the choice of repair pathway at forks stalled within centromeric repeats. Interestingly, the data from DT40 cells showed that PCNA SUMOylation was important to promote template switch upon resolution of replication arrest[9]. Here, we showed that specifically the absence of SUMO chains leads to excessive channeling toward template switch by Rad8-dependent ubiquitination of PCNA. In turn, Rad52 might gain additional access toward repetitive region of centromeres and facilitate enhanced recombination at this locus. Interestingly, a recent report showed that replication machinery inhibited Rad52-mediated single-strand annealing at centromeric repeats[60]. Additionally, we showed that polySUMOylation of Rad52 may itself play an inhibitory role for Rad52 activity at centromeres upon enhanced channeling toward TS in SUMO-KallR. Thus, without SUMO chains, the error-free pathway of TS might become a source of genetic instability.

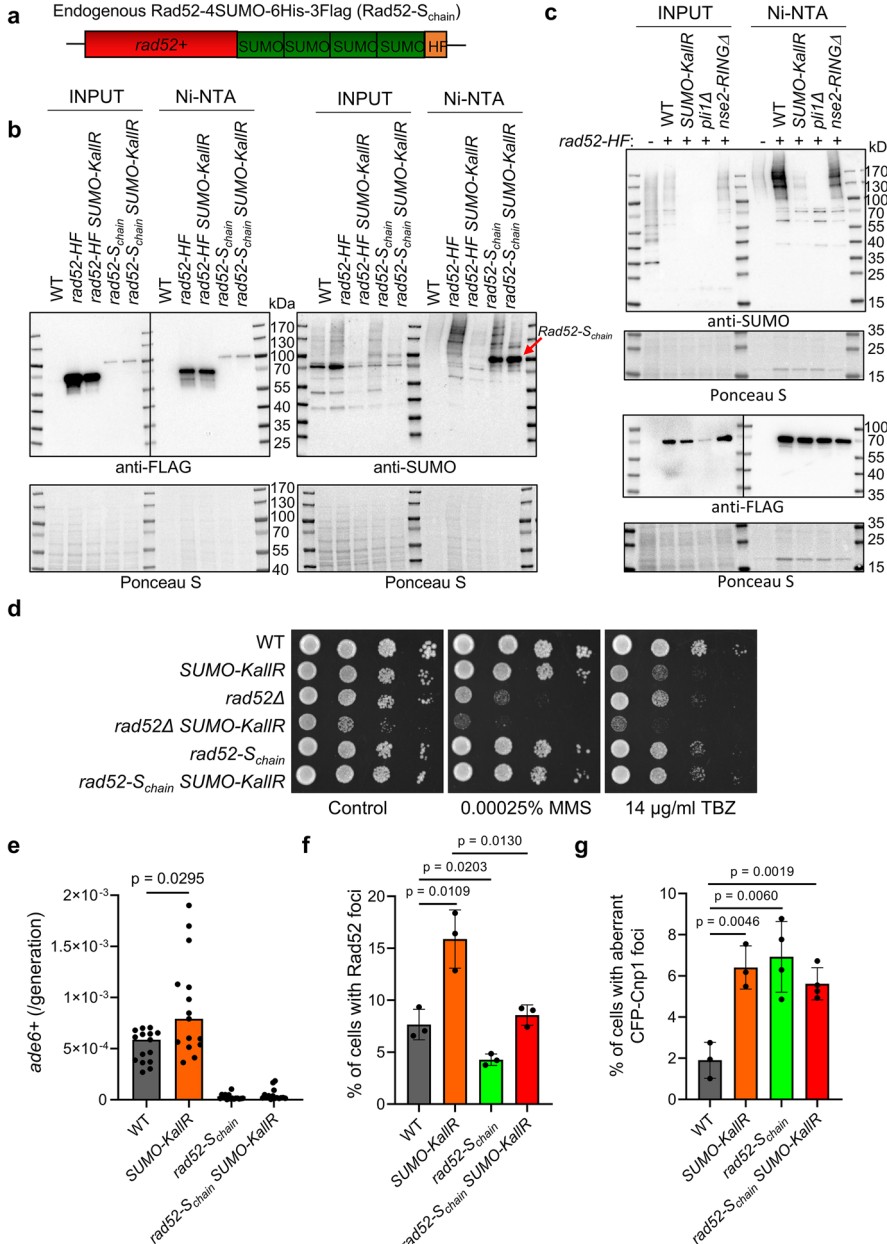

**Fig. 7 | PolySUMOylation of Rad52 restrict centromeric recombination. a** A scheme presenting artificial Rad52-SUMOchain (referred to as Rad52-S$_{chain}$); endogenous *rad52+* gene was tagged with four active (truncated at C-terminus) Pmt3-GG particles followed by 6His-3Flag. **b** SUMOylation status of Rad52-6His-3Flag (Rad52-HF) and Rad52-SUMOchain-6his-3Flag (Rad52-S$_{chain}$) by Ni-NTA-pulldown from indicated strains; WT included as specificity control. Same samples were run to obtain two membranes; left panel was probed against Flag (two exposures are merged, full blots from each exposure presented in Source data), right panel was probed with anti-SUMO antibody. Ponceau S presented to show protein loading. Representative blots from *n* = 3 biological replicates. **c** Endogenous SUMOylation status of Rad52-HF from indicated strains by Ni-NTA pulldown. Same samples were run on separate gels and Rad52-HF precipitation was confirmed by anti-Flag Western blot (bottom panel, two exposure merged, full blots in Source data file). SUMOylation status of precipitated Rad52-HF was examined with anti-SUMO antibody (upper panel). Ponceau S was included for both Western blots to show the amount of protein loaded onto the gels. Representative blots from *n* = 3 biological replicates. **d** Drop dilution assay of indicated strains. Tenfold serial dilution of exponential cultures were dropped on YES agar plates

containing MMS or TBZ in indicated concentrations. **e** Spontaneous rate of centromeric gene conversion as in Fig. 3d, e done on saturated cell cultures of indicated strains. Dots represent values obtained from 15 independent cultures from single colonies. Median is marked for each strain. Two-sided Mann–Whitney *U* test was used to calculate the *p*-values. **f** % of cells forming Rad52 foci based on immunofluorescence images of *radS2-6His-3Flag* (WT), *SUMO-KallR radS2-6His-3Flag*, *rad52-S$_{chain}$-6His-3Flag* and *SUMO-KallR radS2-S$_{chain}$-6His-3Flag* logarithmically growing strains. Fixed cells were probed with anti-Flag (primary) and anti-Rabbit Alexa Fluor 555 (secondary) antibody. Chromatin was stained with DAPI fluorescent dye. Dots represent values obtained from 3 independent biological experiments. At least 600 cells were analyzed for each strain for a single experiment. Error bars show the SD about the mean values. Two-sided Student's *t*-test was used to calculate the *p*-values. **g** % of cells forming aberrant CFP-Cnp1 foci in indicated strains. Dots represent values obtained from 3 independent biological experiments. In all, at least 500 cells were analyzed for each strain. Error bars show the SD about the mean values. Two-sided Student's *t*-test was used to calculate the *p*-values. Source data are provided as a Source data file.

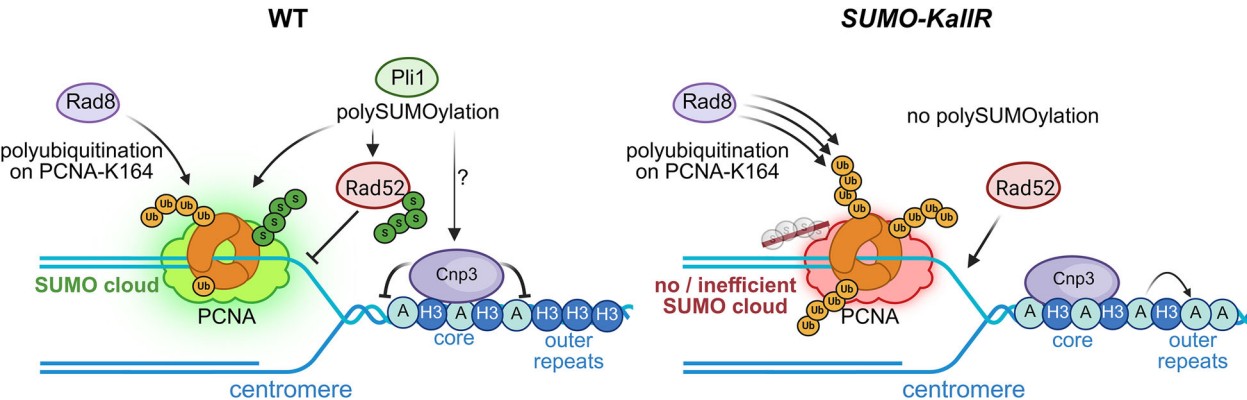

**Fig. 8 | Model depicting impact of SUMO chains on the choice of repair pathway at stalled replication forks within centromeres.** Under physiological conditions, recombination within the centromere is regulated through post-translational modifications of PCNA, including Rad8-dependent polyubiquitination and Pli1-dependent polySUMOylation. In addition, polySUMOylation of Rad52 restricts its access to centromeric regions, thereby further limiting recombination. Accurate deposition of Cnp1 to the centromeric core is dependent on Cnp3 and the cell's ability to assemble SUMO chains. In the *SUMO-KallR* mutant, the absence of PCNA polySUMOylation results in its enhanced polyubiquitination, which leads to increased centromeric recombination. Furthermore, the lack of SUMO chains on Rad52 increases its availability at centromeres, thereby further promoting recombination. In the *SUMO-KallR* mutant, the inhibitory mechanism preventing Cnp1 deposition outside the centromeric core is also abolished. Created in BioRender. Kramarz, K. (2025) https://BioRender.com/rfj0pey.

In fission yeast, SUMOylation of PCNA was so far not certain, in contrast to orthologues from human cells and budding yeast[61]. It was also shown that *S. cerevisiae* PCNA undergo polySUMOylation[62]. In *S. pombe* so far ubiquitination of K164 on PCNA was prevalent and promoted chromatin association of replisome's factors[63]. Here, we present that both monoSUMOylation and SUMO chains might target fission yeast PCNA as well, impacting the choice of post-replication repair pathway at locally stressed forks in unchallenged conditions. Moreover, *pcn1-K164R* allele, as expected, showed almost complete reduction of ubiquitin signal on Ni-NTA pulldown. However, it was followed by an increase in PCNA SUMOylation, in contrast to budding yeast, where K164R mutation on PCNA decreased SUMO attachment to PCNA[62]. Furthermore, when we monitored dynamics of Rad52-YFP foci in response to MMS, the loss of SUMO chains resulted in longer retention of repair proteins compared to WT upon drug removal, either because of discussed above increased TS at the sites of replication arrest or dysregulation in protein turnover at repair centers. We observed an accumulation of RPA foci colocalizing with centromeres upon SUMO chain-deficiency, thus one would speculate that elevated centromeric recombination stems from increased RFs stalling. Genetic analysis showed a suppression of TBZ sensitivity of *SUMO-KallR* mutant by introduction of either *pcn1-K164R* allele or *rad8Δ* deletion, supporting the hypothesis of excessive template switch activation at stalled forks as a cause for elevated centromeric recombination in the absence of SUMO chains. Interestingly, we observed a spreading of the centromeric core histone Cnp1[CENP-A] to *otr* region in *SUMO-KallR*. This was not a consequence of upregulation of Cnp1 levels and we did not observe ectopic binding of Cnp1 by NGS sequencing. CENP-A loading and positioning relies on recombination in human cells[27], so it was possible that in *S. pombe*, analogous mechanisms control centromere maintenance. However, reduction in template switch by *pcn1-K164R* mutation did not alleviate spreading of Cnp1 in *SUMO-KallR* cells. This result suggests that dysregulation of PRR pathways activation upon loss of SUMO chains that leads to excessive centromeric recombination, might be independent of CENP-A loading. The correct localization of Cnp1 is maintained and restricted by the kinetochore network[52]. Deletion of *cnp3+* (hCENP-C) into *SUMO-KallR* cells led to a marked accumulation of Rad52 foci, indicating a strong increase in endogenous DNA damage, potentially arising from mitotic defects. It has though, suppressed Cnp1 spreading in the double mutant *SUMO-KallR cnp3Δ*, without exacerbating declustering of centromeres observed in single *cnp3Δ* cells. Thus, polySUMOylation of kinetochore proteins

might be important for appropriate Cnp1 localization, although this requires further investigation.

## Methods

### Standard yeast genetics

All yeast strains and primers used in this work are listed in Supplementary Tables 1 and 2, respectively. The growth of relevant mutants was done in rich medium (YES) or minimal medium with glutamate as nitrogen source (EMMg). Gene deletion or tagging were performed by classical genetic techniques followed by tetrad dissections to obtain multiple mutants. Reagents for SUMO-ID proteomics were generated by gene synthesis (GenScript) and then subcloned into either pREP41 (in case of 6HA-NTurbo-Pmt3[SUMO]/Pmt3-KallR[SUMO-KallR]) or pFA6a-KanMX6 (in case of 12Pk-CTurbo for Rad52 tagging). The sensitivity of chosen mutants to genotoxic agents was assessed by spotting serially diluted mid log-phase cells onto plates containing HU, MMS, phleomycin (Phleo), camptothecin, or irradiated with UV-C using crosslinker. The silencing of reporter genes in centromeres was determined on EMMg plates without adenine (EMMg−Ade) or uracil (EMMg−Ura) or YES with 5-fluoroorotic acid (5-FOA, Sigma-Aldrich, 47190).

### Live cell imaging

Cells were grown in EMMg medium to exponential phase and dropped onto microscope slide covered with a thin layer of 1.4% agarose resuspended in EMMg. 11 z-stack pictures (each z step of 250 nm) were captured using the Axio Imager M1 epifluorescence microscope (Carl Zeiss) equipped with a 100× immersion oil objective (Plan-Neofluar 1006/1.30), the GFP/dsRED filter set and differential interference contrast (DIC). Images were acquired with AxioCam MRc digital color camera and ZEN 3.7 software (Carl Zeiss). Pictures were processed and counted using ImageJ.

### Rad51 and Rad52-6His-3Flag Immunofluorescence

Logarithmic strains grown in EMMg complete were fixed with 4% formaldehyde, 45 min at room temperature (RT). The excess of formaldehyde was removed by PBS wash and PEM buffer (100 mM PIPES, 1 mM EGTA, 1 mM $MgSO_4$, pH 6.9). Since that moment, all centrifugations were done for 5 min at 600 × *g*. Spheroplasts were obtained by digestion of cells for 10 min at 30 °C with Zymolyase 100 T (BioShop, ZYM002) at concentration 0.5 µg/mL in PEMS (PEM with 1.2 M sorbitol). Zymolyase was removed by three washes with PEMS and then cells were exposed to 1% Triton ×-100 in PEMS for 5 min, RT. Cells were then

washed two times with PEMBAL (1% BSA, 0.1% sodium azide, 100 mM lysine monohydrate (Sigma-Aldrich, L-5626) resuspended in PEM buffer) and then incubated for 1 h on the wheel in PEMBAL, RT. After that time, cells were resuspended in 300 µL of PEMBAL with anti-Rad51 antibody (Abcam, ab63799, 1:300) or anti-Flag antibody for Rad52 detection (Sigma-Aldrich, F7425, 1:300) overnight at 20 °C. Next morning 3 washes with 1 mL of PEMBAL were done—two for 10 min and the last one for 30 min on a wheel, RT. Then cells were resuspended in 300 µL of PEMBAL with anti-Rabbit Alexa Fluor 555 (Invitrogen, A21428, 1:2000) for 1.5 h at RT. After that time, 3 washes with 1 mL of PBS were done, each for 10 min and then cells were resuspended in 1 mL of PBS with DAPI (1:4000) for 3 min. Next, a brief wash with PBS was done to remove the excess of DAPI. Lastly, cells were resuspended in 10 µL of ProLong Gold antifade reagent (Invitrogen, P36934) and subjected to a snapshot microscopy on standard glass slides using Axio Imager M1 microscope (Carl Zeiss). 11 z-stack pictures of 250 nm were collected with 300 ms exposure for Alexa 555 and 20 ms for DAPI channels using ZEN 3.7 software (Carl Zeiss). Images were analysed using ImageJ.

### EdU assay
Yeast cells bearing the human Equilibrative Nucleoside Transporter (hENT1) and the *Drosophila melanogaster* nucleotide (thymidine) kinase (dmNK) were grown in EMMg not supplemented with amino acids and bases until early exponential phase and exposed to a short pulse of the thymidine analog 5-ethynyl-2'- deoxyuridine (EdU) (Invitrogen, A10044) added to each culture to a final concentration of 50 µM for 10 min at RT. Cells were fixed in ethanol 70% and washed once in PBS 1× and twice in 1% Triton-PBS 1×. Cells were resuspended in 200 µL of freshly made "Click-iT" solution: 100 mM sodium ascorbate (from freshly made 1 M stock), 2 mM CuSO$_4$, 0.5 µM Alexa Fluor 488 azide (Invitrogen, A10266) in PBS 1× (components were added in this order to PBS). After 1 h incubation on a wheel at RT in the dark, cells were washed twice in PBS 1×, and next incubated in 1 mL of DAPI-PBS 1× for 15 min. at RT. For imaging, cells were centrifuged, washed in 1 mL of PBS 1× and resuspended in $^1/_{20}$ the original volume. A drop was applied onto a microscope slide, cover with a coverslip and view in fluorescence microscope. Eleven z-stack images (step size of 0.25 µm) were collected using Axio Imager M1 microscope (Carl Zeiss). For EdU-Alexa-488, the samples were excited with a 488 nm laser set at 30% intensity with an exposure time of 100 ms and 20 ms for DAPI channels using ZEN 3.7 software (Carl Zeiss). Images were analyzed with Fiji software. Non-S-phase cells with EdU signal were distinguished by the absence of septum.

### Flow cytometry
Flow cytometry analysis of DNA content was performed as described earlier[64]. Logarithmic cultures were synchronized in early S phase by HU-mediated depletion of the nucleotide pool. After drug removal, cells progressed through S phase and reached G2 phase after 90 min of cycling in fresh medium. At each timepoint, cells were fixed in 70 % ethanol and washed with 50 mM sodium citrate, digested with RNAse A (Sigma-Aldrich, R5503) for 2 h, stained with 1 µM Sytox Green nucleic acid stain (Invitrogen, S7020) and subjected to flow cytometry using Guava easyCyte flow cytometer (Millipore).

### Whole protein extract analysis
Whole protein extraction was performed by standard trichloroacetic acid (TCA) method. 20 mL of logarithmic culture (OD$_{600}$ ~1, $2 \times 10^8$ cells) were collected and resuspended in 300 µl of water. Then, cells were mixed with 350 µl of freshly prepared lysis buffer (2 M NaOH, 7% β-mercaptoethanol) and 350 µl of 50% TCA. After centrifugation, pellets were further washed with 1 M Tris-HCl pH 8 and resuspended in 2× Laemmli buffer (62.5 mM Tris-HCl pH 6.8, 20% glycerol, 2% SDS, 5% β-mercaptoethanol with bromophenol blue). Samples were boiled

before being subjected to SDS-PAGE on Mini-PROTEAN TGX Precast Gel 4–15% (Bio-Rad, 4561086). Western blots were performed with following antibodies at relevant dilutions: 1:1000 anti-GFP (Invitrogen, A11122), 1:3000 anti-HA (Sigma-Aldrich, H6908), 1:3000 anti-PK/V5 (Bio-Rad, MCA1360G), 1:5000 anti-Rad51 (Abcam, ab63799), 1:1000 anti-H3 (Abcam, ab1791), 1:2000 anti-Flag (Sigma-Aldrich, F7425), 1:3000 anti-Rad52 (BioAcademia, 63–003), 1:3000 anti-biotin (Rockland, 200-301-098), 1:500 anti-ubiquitin (P4D1, Santa Cruz Biotechnology, sc-8017) and 1:2000 polyclonal anti-Pmt3$^{SUMO}$ antibodies, raised in rabbit, a gift from Sarah Lambert.

### Ni-NTA pulldown
Logarithmic strains were grown in EMMg without thiamine. 100 mL of culture (OD$_{600}$ ~1) was harvested by centrifugation, washed once with water, and resuspended in 450 µL of buffer G (100 mM sodium phosphate, pH 8.0; 10 mM Tris-HCl, pH 8.0; 6 M guanidine hydrochloride) with 1% Triton ×-100, protease inhibitor cocktail (Sigma-Aldrich, P8215), 1 mM PMSF, and 20 mM N-ethylmaleimide. Cells were lysed by bead beating (4 × 30 s at 6500 rpm with 3 min intervals on ice). Lysates were cleared by centrifugation at $18,000 \times g$ for 20 min at 4 °C, and 20 µL of the supernatant was saved as input. The remaining lysate was applied to Ni-NTA agarose columns (New England Biolabs, S1427L), prewashed with buffer G, and incubated for 1.5 h at 4 °C with rotation. After incubation columns were washed twice with buffer G containing 5 mM imidazole, followed by two washes with buffer U (8 M urea, 100 mM sodium phosphate, pH 6.4; 10 mM Tris-HCl, pH 6.4; 0.1% Triton ×-100) with 5 mM imidazole. Proteins were eluted in two steps by addition of 100 µL of IMAC elution buffer (1× IMAC, 500 mM imidazole) with 5 min incubation per step. Eluates were precipitated with an 200 µL of 50% TCA. Input samples were mixed with 80 µL water and 100 µL 50% TCA. All samples were incubated on ice for 20 min and centrifuged at $18,000 \times g$ for 20 min at 4 °C. Obtained pellets were washed with 1 M Tris-HCl, pH 8.0, centrifuged again, and resuspended in Laemmli buffer (17 µL for IP, 40 µL for input; 62.5 mM Tris-HCl, pH 6.8; 20% glycerol; 2% SDS; 5% β-mercaptoethanol; bromophenol blue). Samples were boiled at 100 °C for 5 min and analyzed by SDS-PAGE.

### Cycloheximide chase
Exponentially growing cells of the wild-type strain and selected mutants were treated or not with cycloheximide to a final concentration of 250 µg/ml. At selected time points, 15 ml of culture at an OD ~ 0.5 was collected, and total protein extracts were prepared using the standard TCA method. Samples were boiled prior to separation by SDS-PAGE using Mini-PROTEAN TGX Precast Gels 4–15% (Bio-Rad, 4561086). Western blotting was performed using anti-GFP (Invitrogen, A11122) and anti-H3 (Abcam, ab1791) antibodies.

### Pulsed field gel electrophoresis
HU-synchronization at the onset of S-phase prevents chromosomes from migrating into the gel due to the accumulation of branched DNA structures. After drug-removal, replication intermediates are resolved, and the migration of chromosomes into the gel is restored. Yeast cultures, grown in YES medium, were diluted to OD$_{600}$ ~ 0.5, then exposed to 20 mM HU for 4 h, washed with water, and released to fresh, drug-free YES medium. At indicated time points 20 mL of culture was collected, washed with cold 50 mM EDTA pH 8 and digested with lyticase (Sigma-Aldrich, L4025) in CSE buffer (20 mM citrate/phosphate pH 5.6, 1.2 M sorbitol, 40 mM EDTA pH 8). Spheroplasts were embedded into 1% UltraPure™ Agarose (Invitrogen, 16500) and put into 4 agarose plugs per each time point. Obtained plugs were incubated in Lysis Buffer 1 (50 mM Tris-HCl pH 7.5, 250 mM EDTA pH 8, 1 % SDS) for 90 min at 55 °C and then digested in Lysis Buffer 2 (1 % N-lauryl sarcosine, 0.5 M EDTA pH 9.5,

0.5 mg/mL proteinase K) overnight at 55 °C. Lysis Buffer 2 was changed the next morning for a fresh one, and digestion was continued overnight at 55 °C. Plugs were then stored at 4 °C. Pulsed field gel electrophoresis was carried out on CHEF-DR-III system (Bio-Rad) for 48 h at 2.0 V/cm, with an angle 120°, at 14 °C. Single switch time was set at 1800s. pump speed 70. Electrophoresis was carried out in 0.8% agarose gel (Agarose MP, Roche, 11388983001) in 1× TAE buffer. Chromosomes were visualized at Chemidoc XRS+ (Bio-Rad) after gel staining in ethidium bromide (EtBr, 10 μg/mL) for 30 min and washing for 30 min in 1× TAE.

### Chromatin immunoprecipitation for standard qPCR analyses

Rad52-YFP and CFP-Cpn1 ChIP were performed as follows. 100 mL of exponential culture ($OD_{600}$ ~ 1) was crosslinked with 10 mM dimethyl adipimidate (Sigma-Aldrich, 285625) for 45 min and subsequently 1% formaldehyde (Sigma-Aldrich, F-8775) for 15 min. Formaldehyde crosslinking was quenched with 2.5 M glycine and cells were washed with 1× TBS. Next, samples were frozen in liquid nitrogen and cells were extracted by bead beating in FA-lysis buffer (50 mM HEPES pH 7.5, 140 mM NaCl, 1% Triton ×-100, 0.1% sodium deoxycholate, 1 mM EDTA with 1 mM PMSF and protease inhibitors (Sigma-Aldrich, P8215)). Chromatin sonication was done in a Bioruptor Pico (Diagenode) using Easy mode 10 cycles of 30 s ON and 30 s OFF. Immunoprecipitation was carried out overnight: 300 μL of extract was incubated on the wheel with anti-GFP antibody (Invitrogen, A11122) at 1:150 concentration. 5 μL of extract was preserved as INPUT fraction. Next morning Protein G Dynabeads (Invitrogen, 10003D) were added for 1 h, stringency washes to remove unspecific proteins were done and immunoprecipitated complexes were decrosslinked for 2 h at 65° C. The DNA associated with precipitated proteins was purified with a QIAquick PCR purification kit (QIAGEN, 28104) and eluted in 300 μL of water. qPCR with SsoAdvanced Universal SYBR® Green Supermix (Bio-Rad, 1725274, primers listed in Table S2) was performed to determine the Ct using CFX Maestro v1.1 (Bio-Rad). Rad52 enrichment was calculated as % of INPUT by subtraction values obtained for the untagged strain. Intergenic locus was presented as internal specificity control. CFP-Cnp1 ChIP-qPCR was calculated as fold enrichment by dividing obtained % of enrichment over values calculated for intergenic locus and then by dividing over the values obtained for untagged control for each pair of primers. Actine was presented as an internal specificity control.

For Rad51 ChIP, cells from logarithmic cultures (100 mL of $OD_{600}$ ~ 1) were formaldehyde crosslinked for 20 min in vivo, then quenched with 2.5 M glycine and frozen in liquid nitrogen. Culture was divided into two equal aliquots of 50 mL and each was extracted by bead beating in 400 μL of FA-lysis buffer. Chromatin sonication was done in a Bioruptor Pico (Diagenode) using Easy mode 10 cycles of 30 s ON and 30 s OFF. Extracts from each aliquot and strain was mixed and immunoprecipitation was carried overnight as follows: 300 μL of extract was incubated with anti-Rad51 antibody (Abcam, ab63799, 1:150), another 300 μL was incubated with Normal Rabbit IgG antibody (Cell Signaling Technology, 2729S, 1:150) on the wheel, 4 °C. 5 μL of extract was preserved as INPUT fraction. Next morning Protein G Dynabeads (Invitrogen, 10003D) were added for 1 h. Then, the beads were washed to remove unspecific proteins and immunoprecipitated complexes were decrosslinked for 2 h at 65 °C. The DNA associated with precipitated Rad51 was purified with a QIAquick PCR purification kit (QIAGEN, 28104) and eluted in 300 μL of water. Rad51 enrichment was calculated based on Ct values and % enrichment. The values obtained for unspecific IgG were subtracted from values obtained for specific Rad51 antibody. Then values calculated for *rad51Δ* strain (a negative control) were subtracted to obtain final, normalized Rad51 enrichment as % of input. Values

calculated for intergenic locus were used as an internal specificity control on graph.

### Chromatin immunoprecipitation of CFP-Cnp1 for next generation sequencing

Logarithmic fission yeast cultures of untagged control, CFP-Cnp1 (WT), and *SUMO-KallR* strain expressing CFP-Cnp1 were mixed with a calibrator—an aliquot of logarithmic budding yeast culture. Immunoprecipitation of Cnp1-bound DNA was then performed. Obtained samples were subjected to next-generation sequencing (NGS) and signals from the IP/INPUT ratio were normalized to the budding yeast calibrator. In details, 100 mL of exponential culture ($OD_{600}$ ~ 1) of untagged strain, WT CFP-Cnp1 and *SUMO-KallR* CFP-Cnp1 were mixed with 10 mL of budding yeast exponential culture Scc1-Pk, used as a calibrator and then crosslinked with 1% formaldehyde (Sigma-Aldrich, F-8775) for 15 min. Formaldehyde crosslinking was quenched with 2.5 M glycine and cells were washed with 1× TBS. Next, samples were frozen in liquid nitrogen and cells were lysed by bead beating in FA-lysis buffer (50 mM HEPES pH 7.5, 140 mM NaCl, 1% Triton ×-100, 0.1% sodium deoxycholate, 1 mM EDTA with 1 mM PMSF and protease inhibitors (Sigma-Aldrich, P8215)). Chromatin sonication was done in a Bioruptor Pico (Diagenode) using Easy mode 10 cycles of 30 s ON and 30 s OFF. Immunoprecipitation was carried out overnight: 300 μL of extract was incubated on the wheel with anti-GFP antibody (Invitrogen, A11122) at 1:150 concentration. 20 μL of extract was preserved as INPUT fraction. Next morning Protein G Dynabeads (Invitrogen, 10003D) were added for 1 h, stringency washes to remove unspecific proteins were done and immunoprecipitated complexes were decrosslinked overnight at 65° C. Next morning samples were digested with RNAse A (Thermo Scientific, EN0531) for 30 min at 37 °C and then with proteinase K at 65 °C for 1 h. The DNA associated with CFP-Cnp1 was precipitated with isopropanol/glycogen/3 M potassium acetate for 1 h at −20 °C. Precipitated DNA was centrifuged for 20 min at 4 °C and then washed with cold 70% ethanol. Finally, DNA pellet was resuspended in 30 μL of Nuclease-Free Water (Sigma-Aldrich, W4502). DNA concentration was determined by Qubit4 Fluorometer (Invitrogen) using Qubit dsDNA HS Assay Kit (Invitrogen, Q32854) and quality control qPCR was done on 10× dilution of obtained DNA suspension with SsoAdvanced Universal SYBR® Green Supermix (Bio-Rad, 1725274, primers listed in Table S2). Samples were then send for DNA libraries preparation and next generation sequencing (Genomed, Warsaw, Poland).

### Chip-seq data analysis

Raw sequencing data was quality controlled by Q30. Low-quality reads and adapter sequences were cut off using Cutadapt[65]. Filtered reads were then mapped against the target genome (*S. pombe* assembly ASM294v2) and against the spike-in genome (*S. cerevisiae* S288C assembly SacCer3) using Bowtie2[66]. Unmapped reads, rDNA regions, mitochondrial DNA, and duplicates were filtered out using Samtools[67]. Samtools was also used to count reads uniquely mapped to ASM294v2 and SacCer3, and these values were used to calculate the occupancy ratio value. Using MACS3[68], we computed the local bias by taking the maximum bias from surrounding 1 kb, 10 kb, the size of fragment length *d*, and the whole genome background. Finally, we combined and generated the maximum background noise and scaled the ChIP and control to the same sequencing depth. We used two different spike-in calibration methods[37,69]. In this manuscript, we present the results obtained from the first calibration method, however, both methods gave us comparable results. Peaks were called using MACS3 *bdgpeakcall* function[68]. In order to visualize mapped reads, bigWig files were created using deepTools[70]. Plots were created around the midpoint of CENs.

## Recombination rates measurement

Single, red colonies from indicated mutants were inoculated into EMMg complete medium for 72 h at 30 °C. Then, cultures were diluted and plated on EMMg−Ade plates with 50 mg/L guanine to block the adenine uptake and score for positive recombination events and EMMg complete plates for cell survival assessment. Plates were incubated for 4–5 days at 30 °C. The rate of recombination was calculated using a fluctuation test. The statistical significance was calculated using two-tailed Mann–Whitney nonparametric test in GraphPad Prism 9.

## *ura4+* transcript level measurements

Total RNA was extracted from exponential cultures (PureLink RNA Mini Kit, Invitrogen, 12183020) and cDNA was prepared (High-Capacity cDNA Reverse Transcription Kit, Applied Biosystems, 4368814) according to the manufacturer's instructions with 1.5 μg of purified RNA. The level of *ura4+* transcript was measured using SsoAdvanced Universal SYBR® Green Supermix (Bio-Rad, 1725274, primers listed in Table S2). Relative level of Ura4 was calculated over actin values.

## Split-SUMO-ID biotinylation assay

Strains with relevant plasmids (empty pREP41 or pREP41-6HA-NTurbo-SUMO) were pre-cultured in 20 mL of EMMg supplemented with biotin and thiamine (to keep the nmt41 promoter from pREP41 shut down). $10^5$ cells were inoculated to 250 mL of EMMg culture (without biotin and thiamine, to induce expression of 6HA-NTurbo-SUMO or 6HA-NTurbo-SUMO-KallR). After 21 h of growth (full induction of nmt41 promoter), 20 mL samples for TCA extraction were collected (-biotin condition), then biotin was added to the final concentration 50 μM and growth was continued for next 3 h. At this point 20 mL of culture was taken for TCA (+ biotin condition). Remaining ~200 mL of induced culture was frozen in liquid nitrogen. Extraction was carried out in RIPA buffer (50 mM Tris-HCl pH 7.5, 150 mM NaCl, 1.5 mM $MgCl_2$, 1 mM EGTA, 0.1% SDS, 1% NP-40, 0.4% sodium deoxycholate, 1 mM DTT, 1 mM PMSF with protease inhibitors (Sigma-Aldrich, P8215) and 20 mM *N*-Ethylmaleimide (NEM, Sigma-Aldrich, 04259) by bead beating. Cell lysate was precleared by centrifugation (20 min, 4 °C, $21130 \times g$). 20 μL of resulting protein extract was recovered and mixed with 4× Laemmli for control Western blot input fraction. Remaining lysates were digested with Benzonase (Sigma-Aldrich, E8263) and RNAse A (Thermo Scientific, EN0531) for 1 h at 4 °C on a rotator wheel. 300 μL of obtained cell lysate was diluted to 1 mL with RIPA buffer, transferred to 2 mL Eppendorf tube and 100 μL of Pierce™ Streptavidin Magnetic Beads (Thermo Scientific, 88817) were added to each sample. Precipitation was carried on a wheel overnight at 4 °C. In the morning, beads were washed thrice with 2 mL of RIPA buffer. Then beads were washed 8 × 2 mL of 10 mM Tris-HCl pH 7.5. Washed beads were resuspended in 100 μL of 10 mM Tris-HCl pH 7.5 and subjected to mass spectrometry analysis.

## Mass spectrometry analysis of Rad52- and SUMO-dependent interactome

On-bead digestion of bound proteins was carried out overnight at 37 °C in the presence of 0.1% RapiGest SF, 3 mM DTT, and 50 ng of trypsin. On the following day, the peptide solution was recovered, and the beads were washed once in 10 mM Tris-HCl pH 7.5, 0.1% RapiGest SF, 3 mM DTT. The flow-through was pooled with the initially recovered solution. The sample was then acidified, high-speed centrifuged, and the supernatant was desalted using the STAGE tip protocol[71]. The obtained peptide pellet was resuspended in a 0.1% formic acid (FA) solution.

LC-MS analyses were carried out with the use of an M-Class Acquity UPLC system connected to a Synapt XS HRMS equipped with a nanoESI ion source interface. Mobile phase A consisted of $H_2O$ + 0.1% FA, while mobile phase B of ACN + 0.1% FA. Trap column (nanoEase C18 100 Å, 5 μm, 180 μm × 20 mm) chromatofocusing was employed before sample separation on the analytical column (nanoEase HSS T3 C18 100 Å, 1.8 μm, 75 μm × 150 mm) with a 60 min linear gradient of 5–35% B at a 300 nL/min flow rate. For each analysis (WT or *SUMO-KallR*), the independent biological replicate count consisted of 3 Split-ID samples (SUMO-ID or SUMO-KallR-ID), and 4 control samples (2 NTurbo-SUMO or NTurbo-SUMO-KallR, and 2 Rad52-CTurbo). In total, 14 samples were analyzed. MS data were collected in ion mobility coupled DIA (HDMS$^E$) at 0.6 scan/s through a 50–2000 m/z range in positive polarity and TOF resolution mode. A collision energy ramp of 27–47 V was set on the instrument's transfer cell. Source conditions: emitter voltage: 2.5 kV, sampling cone: 35 V, source offset: 15 V, source temperature: 80 °C, cone gas flow: 40 L/h, purge gas flow: 0.6 L/h, NanoFlow gas pressure: 0.5 Bar. A (Glu1)-Fibrinopeptide B solution was acquired in the mass reference function, and the correction was applied post-acquisition.

Raw processing was performed using Progenesis QiP v4.2.7. Auto-optimization of the low and high energy ion thresholds was chosen; the LC and TOF peak detection parameters, as well as the peak alignment parameters, were unchanged from QiP's autovalues. Signal intensity within each analysis was normalized across samples using the "Normalize to All proteins" option (a procedure based on robust standard deviation of each samples ions' intensity estimate followed by a scalar multiplication of all ions based on that estimate). Data were searched via Ion Accounting against the *S. pombe* protein sequence databank (UP000002485) to which porcine trypsin, human keratins', BirA, and Streptavidin sequences were appended. For SUMO-KallR-ID, the SUMO gene (*pmt3+*) protein accession was modified, so that all lysines were substituted to arginines. The set search parameters were: peptide mass tolerance: 10 ppm; fragment mass tolerance: 35 ppm; min. fragments/peptide: 1; min. fragments/protein: 2; min. peptides/protein: 1; max. protein mass: 1 MDa; digest reagent: trypsin; max. missed cleavages: 2; variable modification: oxidation of methionine; FDR: 1% protein-level, the estimated peptide-level FDRs were 0.37% and 0.4% for SUMO-ID and SUMO-KallR-ID, respectively. Intensity sum of non-conflicting peptides was used for protein relative quantification.

QiP protein output.csv files were exported and loaded into Perseus[72,73] for statistical work. The protein intensity values were log2 transformed and data normality was confirmed. Only proteins with at least 2 valid values within a group were considered further (26 out of 664, and 20 out of 641 proteins rejected for SUMO-ID and SUMO-KallR-ID, respectively) with any remaining missing values imputed from normal distribution (39 out of 4466, and 53 out of 4347 values for SUMO-ID and SUMO-KallR-ID, respectively). Pearson correlation and principal component analyses were used to assess replicates' reproducibility. Changes in protein quantity were identified with a both-sided *T*-test, assuming $p < 0.05$ (rounded to nearest hundredth; exact cut-off is less than 0.055) and fold-change (FCH) ≥ 1.5 as significance thresholds (logarithmic values: $-log10(p) = 1.25$, $log2(FCH) = 0.585$). Given criteria yielded a total of 96 (70 enriched) and 23 (19 enriched) significantly changed proteins in SUMO-ID and SUMO-KallR-ID, respectively. This data is provided as Supplementary Data 1.

## Statistics and reproducibility

Quantification of chromosomes resolved by PFGE was done with ImageJ and presented as percentage of migrating chromosomes relative to asynchronous profile of chromosomes. Cell imaging was performed with ZEN 3.7 software and images were processed and analysed with ImageJ software. Statistical analysis was carried out using two-sided Student's *t* test for majority of the experiments and

nonparametric Mann–Whitney test for recombination rates analysis. Most experiments were repeated three times, unless otherwise indicated in figure legends.

## Reporting summary

Further information on research design is available in the Nature Portfolio Reporting Summary linked to this article.

## Data availability

The ChIP-seq datasets have been deposited in NCBI's Gene Expression Omnibus and are accessible through GEO Series accession number GSE276805. The mass spectrometry proteomics data have been deposited to the ProteomeXchange Consortium via the PRIDE[74] partner repository with the dataset identifier PXD055556. Reagents and resources are available from corresponding author upon request. Source data are provided with this paper.

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

## Acknowledgements

The authors thank Sarah Lambert, Dorota Dziadkowiec, and Takuro Nakagawa for exchanging reagents, fission yeast strains, and plasmids. We are grateful for valuable discussions with Sarah Lambert and Dorota Dziadkowiec during the course of the project. We would like to thank Silvia Bágeľová Poláková and Robert Wysocki for critical reading of the manuscript. The cnp3Δ and Rad52-6His-3Flag strains were obtained from YGRC/NBRP Japan resource database (http://yeast.nig.ac.jp/yeast/top.xhtml). The purchase of the LC-MS system was financially supported by the "Excellence Initiative—Research University" program for the University of Wrocław. This study was funded by the program "Excellence Initiative - Research University" for the University of Wrocław of the Ministry of Education and Science from Poland, under grant number IDN.CBNDR 0320/2020/20 to K. Kramarz. The funder had no role in study design, data collection and analysis, the decision to publish, or preparation of the manuscript.

## Author contributions

K.M., I.L., J.K., D.M., A.B., M.H., and K.K. performed the experiments. K.M., I.L., and K.K. contributed to experimental design and data analysis. P.T. did bioinformatics analysis of Cnp1 ChIP-seq data. M.T. did mass spectrometry for split-SUMO-ID and bioinformatics analysis. K.K. wrote the manuscript. All authors have read and agreed to the final version of the manuscript.

## Competing interests

The authors declare no competing interests.
