## [Transparent Peer Review file · Nature Communications]

PolySUMOylation of PCNA and Rad52 restricts centromeric recombination in fission yeast

Corresponding Author: Dr Karol Kramarz

Version 0:

Reviewer comments:

Reviewer #2

(Remarks to the Author)

In this manuscript, Markowska et al. investigated the impact of loss of SUMO chains on DNA replication at rDNA and centromere regions in fission yeast *S. pombe*. One interesting observation is that cells deficient in SUMO chain formation exhibit elevated replication stress and DNA damage. Further evidence indicate that SUMO chain formation is required for rDNA stability and the precise centromere localization of CENP-A protein. In addition, SUMO chain-deficient cells show increased replication fork stalling at rDNA region as well as enhanced recombination rate at centromeres. Finally, a split-SUMO-ID proteomic approach was used to identify Rad52-SUMO associated proteins. Although the results from this research indicate the role of polySUMOylation in the replication of the difficult-to-replicate loci (rDNA and centromeres), this manuscript lacks sufficient mechanistic details about the precise role of polySUMOylation in DNA replication at these regions. Major revisions are required before this manuscript can be considered for publication by Nature Communication.

Major points

1. Despite the interesting observations of increased RPA or Rad51/2 foci in rDNA or centromere regions in cells lacking SUMO chains are very interesting, the results in this manuscript are not sufficient to demonstrate how SUMO chains specifically affect replication or nucleosome organization of these repetitive sequences. It is unclear if polySUMOylation functions during normal DNA replication process in these regions or only when DNA damage/replication stress occurs.
2. Previous studies in other organisms demonstrate the SUMOylation of proteins associated with rDNA and centromeres. Identification of the SUMO proteins that directly contribute to rDNA stability and Cnp1/CENP-A centromere localization will strengthen this work.
3. Recent evidence indicates the role of the polySUMOylation in protein delocalization, and the polySUMO-triggered STUbL and Cdc48 complex are required for this delocalization. One possibility is that the lack of SUMO chain in SUMO-KallR mutants contributes to the rDNA stability and aberrant Cnp1 centromere loading through the polySUMO pathway, but this possibility was not tested in this research.
4. Fig. 2h shows efficient Chk1 phosphorylation in SUMO-KallR mutant cells treated with MMS, but it could be more informative to follow Chk1 dephosphorylation kinetics during the recovery from MMS treatment. It is possible that polySUMOylation facilitates checkpoint silencing by disassembling the protein complex at DNA damage foci.
5. It seems that SUMO-KallR mutant cells exhibit increased Cnp1 chromosome association throughout the genome, even though the increase is more dramatic at centromere regions (Fig. 5). Recent studies from the Basrai lab in budding yeast indicate that STUbL Slx5 prevents mislocalization of centromeric CENP-A (Cse4) to chromosome arms by promoting Cse4 degradation. This possibility should be examined in *S. pombe* SUMO-KallR mutants.
6. The authors used Turbo-ID to detect proteins that associate with Rad52-SUMO. While it is a useful tool to study transient protein interactions, the results do not provide sufficient insight into the precise role of Rad52 SUMOylation in the altered rDNA and centromere organization.

Minor points

- Fig. 1b. The decrease of HMW SUMO proteins in SUMO-KallR mutant cells is not very clear. Quantification might help.
- The MW markers are missing in all WB results.
- Several figures include microscopy images for the SUMO-KallR mutant but lack images for the control used. This begins with Figure 1G but is seen throughout the figures.

- Line 97-99: In addition to the possibility of limited recruitment of HR factors in SUMO-KallR mutants, compromised disassembly/delocalization is also possible.
- Line 107: Synthetic sickness? It seems that single SUMO deletion mutant exhibits slow growth, thus this phenotype is not synthetic sickness.
- Fig. 1e/line 120: "...indicating increased spontaneous DNA damage". Another explanation is compromised disassembly of protein complexes at DNA damage foci.
- Fig. 4a/b: The possibility of a kinetochore clustering defect in SUMO-KallR mutant cells needs to be considered.
- Fig. 6e: The authors claim that mutant rad51-3A, showed the most severe sensitivity to TBZ. This is confusing as the most severe defects were seen in rad51 Δ SUMO-KallR and rad52 Δ SUMO-KallR double mutants. The authors should change the wording for clarity.
- Fig. 7b: There is a concern with the anti-HA blot as the streptavidin pulldown of biotinylated proteins shows that HA-SUMO can be non-specifically pulled down in the strain expressing only one subunit of TurboID (HA-NTurbo-SUMO).
- Fig. S4d: It appears that biotin itself can increase protein SUMOylation (WT + HA-NTurbo-SUMO), regardless of split-TurboID reconstitution. The authors should clarify this result.

Reviewer #3

(Remarks to the Author)

The manuscript entitled "SUMO chains fine-tune DNA replication and limit recombination at difficult-to-replicate sites" by Markowska and colleagues reports on the role of SUMO chains in maintaining genome integrity in fission yeast. To prevent the formation of SUMO chains the authors use an allele SUMO-KallR that can be conjugated to lysines but cannot itself be sumoylated. Cells that express SUMO-KallR exhibit a number of phenotypes including increased RPA, Rad51 and Rad52 foci, and increased instability of the rDNA loci, which is accompanied by replication stalling in the rDNA and Rad52 foci in the nucleolus. Furthermore, other hard-to-replicate loci such as the centromeres are also destabilized by the SUMO-KallR allele. To some extent the SUMO-KallR appears to be a hypomorph with an intermediate phenotype of WT and SUMO Δ . Finally, the authors have developed a split-SUMO-ID method, which they use to identify protein in the proximity of Rad52, which are sumoylated or bind to sumoylated proteins. The manuscript is well-written and the data are of high quality, but unfortunately the study does not extend much from previous studies in terms of elucidating the specific role of SUMO chains and none of the hits from the split-SUMO-ID screen are validated and characterized. For specific suggestions and recommendations, see below:

Major issues to address

1. Line 126: the authors cannot conclude from figure 1 that replication stress is reduced by SUMO-chains, because replication stress was not examined in figure 1.
2. Line 134: the altered progression through S phase after release from HU arrest is not convincing in Fig. S2c. The claim should be supported by additional data.
3. Figure 3d reports the % colocalization rather than the percentage of cells with each type of foci, which could be misleading given that the number of cells with foci also changes.
4. A recent study identified poly-sumoylation as a regulator of rDNA replication in budding yeast (Gutierrez-Morton et al., 2024). The potential link to the current study should be discussed.
5. Sumoylation has been linked to liquid-liquid phase separation. Might the aberrant Cnp1 localization reflect a defect in LLPS at centromeres?
6. The split-TurboID is not tied in well with the rest of the manuscript and no validation or characterization of hits is performed e.g. actin. Can the authors observe interactions that are specific for SUMO chains using 6HA-NTurbo-SUMO-KallR?
7. Several studies have analyzed SUMO chains in budding yeast e.g. (Bylebyl et al., 2003; Newman et al., 2017; Psakhye et al., 2019). The results of the current study should be compared in greater detail to previous studies.

Minor suggestions

1. Line 49: change "rearrangements" to "deletions".
2. The manuscript could benefit from a systematic correction of English grammar.
3. Line 176: the authors should be more specific about the observed change to chromosome III. To me, the band simply looks more smeary.
4. Line 259: replace "interacted" with "colocalized".
5. Line 265: it is unclear what comparison "most severe" refers to. The double mutants are more sensitive to TBZ than rad51-3A.
6. It is not entirely clear from the methods and legend what is shown in figure 7C. If I understand correctly it is the ratio of biotinylated proteins in pREP41 or pREP41-6HA-NTurbo-SUMO strains expressing Rad52-12PK-CTurbo in untreated conditions.

References:

- Bylebyl, G.R., I. Belichenko, and E.S. Johnson. 2003. The SUMO isopeptidase Ulp2 prevents accumulation of SUMO chains in yeast. *J Biol Chem.* 278:44113-44120.
- Gutierrez-Morton, E., C. Haluska, L. Collins, R. Rizkallah, R.J. Tomko, Jr., and Y. Wang. 2024. The polySUMOylation axis promotes nucleolar release of Tof2 for mitotic exit. *Cell reports.* 43:114492.
- Newman, H.A., P.B. Meluh, J. Lu, J. Vidal, C. Carson, E. Lagesse, J.J. Gray, J.D. Boeke, and M.J. Matunis. 2017. A high throughput mutagenic analysis of yeast sumo structure and function. *PLoS Genet.* 13:e1006612.
- Psakhye, I., F. Castellucci, and D. Brnzei. 2019. SUMO-Chain-Regulated Proteasomal Degradation Timing Exemplified in DNA Replication Initiation. *Mol Cell.* 76:632-645 e636.

Version 1:

Reviewer comments:

Reviewer #2

(Remarks to the Author)

The authors have done an excellent job to address all the concerns. I believe this revised version is suitable for publication after some minor issues are fixed.

Line 179: Fig. S4a-b: should be Fig. S5a-b.

Fig. 6e: the temperature (34°C) for the red group experiment needs to be labelled.

For the discussion section, several paragraphs are too long.

Reviewer #3

(Remarks to the Author)

The revised manuscript NCOMMS-24-57672A by Markowska and colleagues has addressed many of my concerns. The most striking finding of the manuscript, is the complete suppression of TBZ sensitivity of SMT3-KallR by rad8Δ or Rad52-Schain. However, the revised manuscript raises a number of new questions that are detailed below.

Major issues to address

1. Are Rad52 and PCNA separate sumo pathways? The authors suggest so, but I am not convinced. Additional data to resolve this question would strengthen the manuscript.
2. Line 184-185: the spreading of Cnp1CENP-A to outer repetitive sequences of centromere reported in figure 3g for the KallR mutant is not convincingly different from WT.
3. Line 206: the increased binding of Rad52 and Rad51 to chromatin does not appear to be specific for centromeres, which contradicts the conclusion in lines 223-225 that SUMO chains limit recombination at centromeres, as opposed to chromosome arms, where they seem to promote recombination.
4. Line 273: if K164 is not the sumoylation site of Pcn1 is it then K127 as in budding yeast?
5. Fig 7e: It is also important to know the percentage of non-colocalizing Rad52 foci to know if the hyperrecombination is specific to centromeres.
6. A complete list of proteins identified in the SUMO-ID screen (Fig. 4c-d) should be made available.
7. The notion that SUMO chains attachment to Rad52 safeguards centromeres is based on the Rad52-Schain construct, which may be a hypomorph allele (Fig 8f, reduced foci). It would strengthen the conclusion, if the sumoylation sites in Rad52 could be mutated and the phenotype analyzed. The prediction is that a sumoylation-defective Rad52 should be hyperrecombinant for centromere recombination.
8. Fig s3c: repair in mutant is delayed 4 hours, but kinetics is similar to wt? Is that because an alternative repair pathway is engaged after a delay?
9. The finding from the RAD52-SUMO-ID that Rad52 leads to biotinylation of certain proteins (Pcn1, Pol3, etc), when they are polysumoylation, raises the question if any of these proteins interact directly with Rad52.

Minor suggestions

1. Line 130: change "pulse-field" to "pulsed-field".
2. Line 157-159, 204: change "2a" to "3a" and "2b" to "3b".
3. Line 163: change "2c" to "3c".
4. The order of panels in figures 2 and 3 do not follow the text.
5. Line 182: change "2g" to "3g".
6. Line 211 and elsewhere: change "3c" to "2c".
7. Line 234: please rephrase the sentence "The Rad52-12Pk-Cturbo strain resembled WT phenotypes to genotoxic agents and phenocopied SUMO-KallR in contrast to rad52Δ".
8. Line 277 and elsewhere: change "6a" to "7a".
9. Line 297 and elsewhere: change "7b" to "8b".

Reviewer #4

(Remarks to the Author)

The manuscript entitled "PolySUMOylation of PCNA and Rad52 restricts centromeric recombination in fission yeast" by Markowska et al. investigates the effects of preventing SUMO chain formation on DNA repair and centromere integrity in *Schizosaccharomyces pombe*. Using a SUMO variant Pmt3-KallRSUMO-KallR that allows only monoSUMOylation, the authors show that SUMO-chain-deficient cells accumulate DNA damage under replication stress and mislocalize the centromere-specific histone Cnp1. They further study the Slx8 STUbL pathway in centromere maintenance and adapt a Split-SUMO-ID proteomics approach by comparing Pmt3SUMO strain with Pmt3-KallRSUMO-KallR strain to identify SUMO-dependent Rad52 interactors, focusing on Rad52 and PCNA to generate a mechanistic model. The work is well written and of potential interest. However, the proteomics analysis and mechanistic evidence require significant strengthening before

publication. Detailed major and minor comments are provided below.

Major Points

1. The label-free quantification (LFQ) experiments were performed with only two replicates per condition (May 2024 and March 2025). This is insufficient for robust statistical analysis; typically at least three independent biological replicates are required for reliable quantification. The manuscript does not clarify whether the two replicates are biological or technical. The authors should include at least three independent biological replicates of the LC-MS/MS experiments and perform reproducibility analyses (e.g. principal component analysis and Pearson correlation of replicates) to demonstrate data quality and consistency.
2. The Split-SUMO-ID approach, as implemented here, has two key drawbacks:
Lack of direct site identification: The current strategy does not identify the lysine residues that are SUMOylated. To map SUMOylation sites, the authors could use a Pmt3-L109R mutant SUMO, which leaves a di-glycine tag on modified peptides after trypsin digestion. This would allow MS-based identification of the exact SUMOylation sites.
SIM-binding ambiguity: Identified interactors may bind to the SUMO moiety via SUMO-interacting motifs (SIMs) rather than being directly SUMOylated. In other words, some hits could be SIM-dependent binding partners instead of polySUMOylated substrates. To resolve this, the authors should include additional controls. For example, a SUMO Δ GG (conjugation-deficient) construct and a Rad52 Δ SIM mutant would help distinguish direct SUMOylation targets from SIM-mediated interactors.
3. Lines 675–677 describe applying different false discovery rate (FDR) cutoffs at various levels, but the rationale is unclear. For high-confidence identifications, a consistent 1% FDR at the peptide and protein levels is standard. The authors should clarify and justify their FDR criteria, and ideally use a consistent 1% FDR for proteins, peptides, and any modifications to ensure comparability.
4. The statistical analysis reports p-values < 0.05 for significance, but no multiple hypothesis testing correction is mentioned. Given the large number of proteins tested, an appropriate correction (e.g. permutation-based FDR, Benjamini-Hochberg) should be applied to control false positives. The authors should specify how p-values were calculated and adjust for multiple comparisons.
5. Four SUMOylation sites on Rad52 have been reported in *S. cerevisiae* (two in the N-terminal region, one in the central domain, one in the C-terminal region). The exact number of SUMO moieties in the polySUMO chain is not known. In Figure 7a, the authors fuse four SUMOs to the C-terminus of Rad52, which appears to mimic the four monoSUMOylation sites. However, since the reported SUMOylation sites are primarily located near the N-terminus, the authors should elaborate on the rationale for choosing a C-terminal fusion. If the goal is to simulate endogenous SUMOylation, it may be more appropriate to fuse SUMOs to the N-terminus. The rationale for using a 4xSUMO fusion other than 3 or 5 should also be explained, and an N-terminal fusion should be tested to confirm that the construct functions similarly.
6. The western blot in Figure 7c intended to validate the Rad52-4SUMO fusion is unclear. Additional controls are needed. For example, the authors could express Rad52-HF in wild-type, Pmt3SUMO, and Pmt3-KallR backgrounds, perform Ni-NTA pulldown, and probe the blots with both anti-Flag and anti-SUMO antibodies. Running the Rad52-4SUMO-HF sample alongside these controls on the same blot would allow direct comparison of band patterns and confirm the presence of polySUMOylated Rad52 species.
7. Given existing knowledge of SUMOylation in yeast proteins, the authors should identify which lysines are SUMOylated in their system. This could be done by site-directed mutagenesis of candidate lysine residues (followed by western blot to test loss of SUMOylation) or by MS analysis using the Pmt3-L109R strain to detect di-glycine remnants on modified peptides. Mapping the SUMO sites on Rad52 (and other key proteins) would strengthen the mechanistic conclusions.
8. While Pli1 is identified as mediating PCNA polySUMOylation, it is important to determine which SUMO E3 ligase is responsible for Rad52 polySUMOylation. The authors should use a similar approach (e.g. testing Rad52 SUMOylation in mutants of known SUMO E3s) to identify the ligase(s) involved. This would add mechanistic detail to the model of how SUMO chains on Rad52 are generated.
9. The authors observe a strong decrease in proteins identified by Rad52-SUMO-ID in the Pmt3-KallR strain (lines 351–354), which contrasts with a prior Molecular Cell study. A plausible explanation is that without SUMO chains on Rad52, many SIM-dependent interactors no longer bind. This implies that several hits in the SUMO-ID may be proteins that interact via SUMO–SIM binding, rather than being PolySUMOylated themselves. The authors should discuss this possibility. It could mean that some identified “interactors” in the wild-type strain are indirect (SIM-binding), and their loss in the KallR strain explains the reduced list. Explicitly considering which hits are truly SUMOylated versus SUMO-binding is important.
10. The schematic model in Figure 8 should be clarified. Currently, it is not clear which additional site of PCNA and site of Rad52 are SUMOylated on each side of the model. The authors should explicitly mark the SUMOylated substrates in the diagram. In particular, including the roles of Cnp1 and Cnp3 (and their SUMOylation status, if known) would provide a more complete picture of the centromere maintenance mechanism. Additional annotation or explanation in the figure legend might help convey how these factors interact in the proposed model.

Minor Points

1. In lines 23–25 the authors refer to “site-specific SUMOylation,” but only the single known site K164 of PCNA is mutated in the study. This wording is misleading unless additional SUMOylation sites have been systematically identified. The authors should clarify or revise the phrasing (e.g. specify “SUMOylation at PCNA-K164” or remove “site-specific” if only one site is analyzed).
2. In lines 343–344, the term “SUMOylome” is used. This suggests a comprehensive list of SUMOylated proteins, but the Split-SUMO-ID dataset likely includes proteins that bind via SIMs and are not directly SUMOylated. To avoid confusion, a more precise term like “SUMO-dependent interactome” or an explicit explanation would be better.

Version 2:

Reviewer comments:

Reviewer #3

(Remarks to the Author)

The revised manuscript NCOMMS-24-57672B by Markowska and colleagues has addressed all of my remaining questions.

Reviewer #4

(Remarks to the Author)

The authors have addressed most of the previous concerns in a satisfactory manner. Before the manuscript can be considered for publication, a few issues still require clarification:

1. Statistical threshold inconsistency (Lines 1007–1008): the manuscript states that significance thresholds were set at “ $p \leq 0.05$ and fold-change (FCH) ≥ 1.5 (corresponding to $-\log_{10}(p) = 1.25$, $\log_2(\text{FCH}) = 0.585$)”. However, $-\log_{10}(p = 0.05)$ equals 1.301, not 1.25. Applying the correct cutoff would exclude additional protein hits currently listed in the supplementary tables SplitID_SUMO_WT_volc and SplitID_SUMO_KallR_volc. Please correct this inconsistency and update the results accordingly.
2. Discrepancy in Perseus output: the “+” hits in column C (“Student’s T-test Significant SplitID_Controls”) of the supplementary spreadsheets are not consistent with the results in the uploaded Perseus project file (250331_CBNDK_KKR_SPLIT_ID.sps). The authors should clarify this discrepancy and ensure that the reported thresholds and lists of significant hits are fully aligned with the original analysis files.

REVIEWER COMMENTS

Reviewer #2 (Remarks to the Author):

The authors have done an excellent job to address all the concerns. I believe this revised version is suitable for publication after some minor issues are fixed.

We are very grateful for the positive judgment. All the suggested comments are incorporated.

Line 179: Fig. S4a-b: should be Fig. S5a-b.

Corrected.

Fig. 6e: the temperature (34°C) for the red group experiment needs to be labelled.

Corrected.

For the discussion section, several paragraphs are too long.

Certain parts of Discussion section were rewritten.

Reviewer #3 (Remarks to the Author):

The revised manuscript NCOMMS-24-57672A by Markowska and colleagues has addressed many of my concerns. The most striking finding of the manuscript, is the complete suppression of TBZ sensitivity of SMT3-KallR by *rad8Δ* or Rad52-Schain. However, the revised manuscript raises a number of new questions that are detailed below.

We thank the Reviewer for evaluating our work. Below, we address the specific comments and rebuttals to the new questions raised.

Major issues to address

1. Are Rad52 and PCNA separate sumo pathways? The authors suggest so, but I am not convinced. Additional data to resolve this question would strengthen the manuscript.

To address this interesting concern, we first examined the genetic interaction between *rad52Δ* and *pcn1-K164R*, and found that the double mutant is lethal (Fig. S14a). Next, we constructed double and triple mutants regarding *SUMO-KallR*, *rad52-Schain* and *pcn1-K164R*. We found a strong negative genetic interaction between *Rad52-Schain* and *pcn1-K164R*, when compared double mutant to each single on MMS and TBZ plates (Fig. S14b). Introduction of *SUMO-KallR* allele was not exacerbating the phenotype of double *rad52-Schain pcn1-K164R* on TBZ plates (Fig. S14c). It underlies that these pathways are independent at least in terms of replication stress and centromere maintenance and critical for cell survival.

2. Line 184-185: the spreading of Cnp1CENP-A to outer repetitive sequences of centromere reported in figure 3g for the KallR mutant is not convincingly different from WT.

The calculation of ChIPseq was done in the most stringent way and all presented enrichment is significant. We are grateful for this comment as in the previous version of manuscript we did not properly describe the Y axis on presented graphs. Now it clearly states that the signal on Y-axis is presented as the $-\log_{10}(\text{p-value})$.

We appreciate the Reviewer's comment and acknowledge that in the genome-wide view (Supplementary Fig. S6) or +/- 50kb window around centromeres (Fig. 2) the centromeric mega-peak ($-\log_{10}$ p-value ~ 4000) masks the subtle pericentromeric differences. To address this, we have generated additional figures that focus exclusively on the pericentromeric regions (within +/-50 kb window around centromeres) with an adjusted y-axis range (see Fig. 2g and Fig. S5c-d). Our analysis shows that the $-\log_{10}$ (p-values) from peak calling using MACS3 (after correcting for multiple testing as described in Materials and methods) exhibit an increase of approximately 200 units in the pericentromeric regions of the *SUMO-KallR_CFP-cnp1* compared with the untagged sample. We believe this focused visualization clarifies the biological importance and addresses the Reviewer's concern effectively.

3. Line 206: the increased binding of Rad52 and Rad51 to chromatin does not appear to be specific for centromeres, which contradicts the conclusion in lines 223-225 that SUMO chains limit recombination at centromeres, as opposed to chromosome arms, where they seem to promote recombination.

We are sorry for not underlining in a comprehensive way, that at intergenic loci, in both, Rad52 and Rad51 ChIP the difference observed is not statistically significant. We now added ns at both intergenic loci (Fig. 3a-b). Thus, observed enrichment is indeed specific for centromeres. Overall, we show global increase of Rad52 or Rad51 by microscopy approaches in *SUMO-KallR* mutant and their local enrichment at centromeres by ChIP. We also showed increased colocalization between Rad52 and Sad1 in *SUMO-KallR* background. Importantly, the recombination rates are increased only at centromeres but not on *ade6/his3* reporter on chromosome arm. Together with data on *pcn1-K164R*, we hypothesize that because replication forks are not frequently stalled at random arm locus, SUMO chains might differently contribute to recombination process.

4. Line 273: if K164 is not the sumoylation site of Pcn1 is it then K127 as in budding yeast?

Investigating the SUMOylation sites on PCNA is an interesting point to address, which will require additional work, including identification of the modified sites by MS and further characterisation of particular lysine mutation on PCNA SUMOylation, followed by functional analysis. Given the in vitro data presented in the PhD thesis by Lauren Small from Felicity Watts laboratory in Sussex University, there are potentially four SUMOylation sites – K13, K164, K172, K253. This resembles SUMOylation of human PCNA. Further studies are required to confirm those sites, as the data from PhD work were not published as a scientific manuscript. We believe that addressing this point is out of the scope of this manuscript and not investigating it in the frame of the revision process does not alter the main message and conclusions of our manuscript.

5. Fig 7e: It is also important to know the percentage of non-colocalizing Rad52 foci to know if the hyperrecombination is specific to centromeres.

Fig7e refers to centromeric recombination measurements and does not address the percentage of Rad52 foci. It shows abolished centromeric recombination upon introduction of a SUMO chain on Rad52. As presented on Fig 3g, we have monitored recombination outside centromeres and detected no hyper-recombination in *SUMO-KallR* mutant (Fig 3f-g). We also present evidence that Rad52 more frequently associates with centromeric locus, both by microscopy and chromatin immunoprecipitation. Thus, we suggest that observed hyperrecombination is specific to centromeres.

6. A complete list of proteins identified in the SUMO-ID screen (Fig. 4c-d) should be made available.

A list of proteins identified in the SUMO-ID screen is presented as Supplementary Dataset.

The full protein lists, with all identification details, are available through the repository files. These are currently only accessible to the Authors and Reviewers (via the Reviewer token submitted to the Editor). A direct link to repository with author login is available through reporting summary file. Relevant information are also present in Data Availability section. As soon as the manuscript is published all data in the repository will be public.

Based on the criticism raised by the Reviewer 4, we replaced the datasets with a complete experiment with three independent biological replicates for WT and *SUMO-KallR*, which was done in March 2025. In the previous version of the manuscript, we kept the original WT samples (done in May 2024) and reanalyzed it alongside *SUMO-KallR* samples (obtained in March 2025) using the same pipeline to generate the volcano plots and compare WT with *SUMO-KallR* identifications.

Given the criticism raised by the Reviewer 4, particularly strong requirement regarding 3 biological replicates for correct statistical analysis, we replaced initial datasets with the complete experiment with samples from WT prepared in parallel with *SUMO-KallR*. Thus, the final volcano plots show differences compared to initial graphs, but most importantly PCNA and Pol3 are significantly enriched only in the WT samples and detected, but not statistically significant in *SUMO-KallR* background. Thus, despite some differences compared to initial WT dataset from May 2024, our main conclusions remain valid, regarding identification of PCNA and claim that lack of SUMO chains led to a vast reduction of proteins identified at the sites of Rad52 repair.

7. The notion that SUMO chains attachment to Rad52 safeguards centromeres is based on the Rad52-Schain construct, which may be a hypomorph allele (Fig 8f, reduced foci). It would strengthen the conclusion, if the sumoylation sites in Rad52 could be mutated and the phenotype analyzed. The prediction is that a sumoylation-defective Rad52 should be hyperrecombinant for centromere recombination.

Our strategy allows us to distinguish between mono versus polySUMOylation. We have directly tested the effect of a SUMO chain presence on Rad52 activity. Abolishing all SUMOylation sites on Rad52 will lead to a complete loss of SUMOylation on the protein, which might seriously affect its activity and localisation. Besides, there are 35 lysines in *S. pombe* Rad52 protein. Abolishment of all SUMOylation sites will greatly affect the overall protein structure, leading even to its misfolding. Based on bioinformatics tools that allow to predict SUMOylation sites, at least 5 up to 8 potential SUMOylation sites are recognised. Generation of relevant rad52-X_KR mutant will consume significant time and falls beyond the scope of the current manuscript. Please see our answer for point 5 of Reviewer 4.

In regard to the Reviewer's prediction, our results indicate that the absence of poly-SUMOylation leads to increased recombination within centromeres. However, it remains unclear whether this increase arises from elevated monoSUMOylation or solely from the lack of polySUMOylation. Based on our genetic analysis of *SUMO-KallR* mutant, in which Rad52 is not polySUMOylated but may still be monoSUMOylated, we propose that monoSUMOylation of Rad52 drives hyper-recombination at the centromere, whereas a complete loss of SUMO attachment to Rad52 might result in distinct phenotypes.

8. Fig s3c: repair in mutant is delayed 4 hours, but kinetics is similar to wt? Is that because an alternative repair pathway is engaged after a delay?

We hypothesize that the alternative recombination pathway, upregulated upon SUMO chain loss, might be the template switch, given strong genetic interaction between *SUMO-KallR* and *pcn1-K164R*. Data obtained for centromeric locus regarding possible overstimulation of template switch supports prolonged Rad52 association with chromatin visualized by microscopy.

9. The finding from the RAD52-SUMO-ID that Rad52 leads to biotinylation of certain proteins (Pcn1, Pol3, etc), when they are polysumoylated, raises the question if any of these proteins interact directly with Rad52.

The point raised by the Reviewer is insightful, and elucidating the direct interactome of Rad52 upon SUMOylation will be an important direction for future research. However, in the current manuscript, our objective was to decipher the interactome landscape (including direct and indirect protein-protein interactions) of Rad52 protein dependent of SUMOylation. The SUMO-ID assay is based on proximity labelling. It is supposed to detect and identify a 'SUMO cloud' at the sites of Rad52 repair, not whole Rad52-interactome. Both, Rad52 and SUMO are significantly enriched in WT and *SUMO-KallR* samples. Establishing which proteins interact directly or not with Rad52 will be a valuable set of information, but falls beyond the scope of current manuscript and that does not alter the overall conclusion of the manuscript.

Minor suggestions

1. Line 130: change "pulse-field" to "pulsed-field".

Corrected

2. Line 157-159, 204: change "2a" to "3a" and "2b" to "3b".

3. Line 163: change "2c" to "3c".

4. The order of panels in figures 2 and 3 do not follow the text.

5. Line 182: change "2g" to "3g".

6. Line 211 and elsewhere: change "3c" to "2c".

Points 2-6: the order seems to be correct, we are sure that the text follows the panels in Figures 2 and 3 correctly.

7. Line 234: please rephrase the sentence "The Rad52-12Pk-Cturbo strain resembled WT phenotypes to genotoxic agents and phenocopied SUMO-KallR in contrast to *rad52Δ*".

Rephrased as follows: "The Rad52-12Pk-CTurbo strain exhibited no apparent sensitivity to genotoxic agents and mirrored the *SUMO-KallR* phenotypes when introduced to this background, in contrast to highly sensitive *rad52Δ* mutant (Fig. S9b-c)."

8. Line 277 and elsewhere: change "6a" to "7a".

9. Line 297 and elsewhere: change "7b" to "8b".

Points 8-9: the order is correct, we are sure that the text follows the panels in Figures 6 and 7 correctly.

Reviewer #4 (Remarks to the Author):

The manuscript entitled “PolySUMOylation of PCNA and Rad52 restricts centromeric recombination in fission yeast” by Markowska et al. investigates the effects of preventing SUMO chain formation on DNA repair and centromere integrity in *Schizosaccharomyces pombe*. Using a SUMO variant Pmt3-KallRSUMO-KallR that allows only monoSUMOylation, the authors show that SUMO-chain-deficient cells accumulate DNA damage under replication stress and mislocalize the centromere-specific histone Cnp1. They further study the Slx8 STUbL pathway in centromere maintenance and adapt a Split-SUMO-ID proteomics approach by comparing Pmt3SUMO strain with Pmt3-KallRSUMO-KallR strain to identify SUMO-dependent Rad52 interactors, focusing on Rad52 and PCNA to generate a mechanistic model. The work is well written and of potential interest. However, the proteomics analysis and mechanistic evidence require significant strengthening before publication. Detailed major and minor comments are provided below.

Major Points

1. The label-free quantification (LFQ) experiments were performed with only two replicates per condition (May 2024 and March 2025). This is insufficient for robust statistical analysis; typically at least three independent biological replicates are required for reliable quantification. The manuscript does not clarify whether the two replicates are biological or technical. The authors should include at least three independent biological replicates of the LC-MS/MS experiments and perform reproducibility analyses (e.g. principal component analysis and Pearson correlation of replicates) to demonstrate data quality and consistency.

Our initial experiment (May 2024) done for WT samples was based on a comparison of 2 studied samples vs 4 background controls. The experimental design of the SPLIT-ID technique determined a higher number of background controls, than studied samples themselves, as each positive SPLIT-ID sample was accompanied by two negative controls (large CTurbo fused to Rad52 and small NTurbo fused to SUMO, the inactive subunits of Turbo ligase). The samples were obtained from two independent biological replicates. We believed this constituted quite a firm background elimination and was even more stringent to the typical “3 samples vs 3 controls” design for a traditional proximity labeling assay.

However, given that the criticism raised by the Reviewer starts with underlining the need to use three replicates as a determinant for reliable quantification, we decided to replace the initial datasets presented in the former version of our manuscript.

In the present version, we provided a full, complete dataset generated in March 2025, where *SUMO-KallR* samples were accompanied by WT samples, performed on three independent biological replicates for positive SPLIT-ID versus four negative background controls. This is now clearly stated in the Figure legend. Despite some differences compared to initial WT dataset from May 2024, our main conclusions remain valid. We do observe a vast decrease in detected proteins from *SUMO-KallR* background compared to WT. Most important, we identified replication factors including Pcn1 (PCNA) and Pol3 in this SUMO-ID experiments, which reached statistical significance only in WT samples.

In the revised manuscript, according to the suggestion of Reviewer we now present reproducibility analyses (PCA and Pearson correlation of replicates) as Supplementary Figure S11. These were also performed earlier, as they are a part of the QiP software’s analytical workflow, but were indeed not presented. For further transparency we now performed these analyses in a widely known and open proteomic environment, Perseus, and

we uploaded the Perseus project file to the repository for viewing. Proper changes were made in the methodological section, and repository files are updated.

2. The Split-SUMO-ID approach, as implemented here, has two key drawbacks:

Lack of direct site identification: The current strategy does not identify the lysine residues that are SUMOylated. To map SUMOylation sites, the authors could use a Pmt3-L109R mutant SUMO, which leaves a di-glycine tag on modified peptides after trypsin digestion. This would allow MS-based identification of the exact SUMOylation sites.

SIM-binding ambiguity: Identified interactors may bind to the SUMO moiety via SUMO-interacting motifs (SIMs) rather than being directly SUMOylated. In other words, some hits could be SIM-dependent binding partners instead of polySUMOylated substrates. To resolve this, the authors should include additional controls. For example, a SUMO Δ GG (conjugation-deficient) construct and a Rad52 Δ SIM mutant would help distinguish direct SUMOylation targets from SIM-mediated interactors.

The point raised by the Reviewer is interesting and deciphering the direct SUMOylation sites on detected proteins would be an exciting research direction. However, in the current manuscript, our objective was to decipher the overall interactome landscape (including direct and indirect protein-protein interaction) on Rad52 protein in dependence of SUMOylation. The SUMO-ID assay is based on proximity labelling. It is supposed to detect and identify a SUMO-dependent interactome (potential SUMO cloud) at the sites of Rad52, not directly a SUMOylome.

We did not attempt to present another SUMOylome list – it has been already done by Thon lab (Køhler, J., Tammsalu, T., Jørgensen, M. *et al.* Targeting of SUMO substrates to a Cdc48–Ufd1–Npl4 segregase and STUbL pathway in fission yeast. *Nat Commun* **6**, 8827 (2015); doi.org/10.1038/ncomms9827).

Our goal was to design a tool providing insight into particular Rad52-mediated DNA repair centers. Moreover, Rad52 in *S. pombe* has no published SIMs or SUMOylation sites (it contains 35 lysines in its sequence, bioinformatics analysis suggests several lysines in the core and C-terminus as potential SUMOylation sites). Predicting and identifying SIMs or SUMO sites on Rad52 is out of scope of this manuscript. Please see below in the answer for comment #5 a scheme with marked potential SUMOylation sites and SUMO-interacting motifs within Rad52.

On the other hand, introduction of Pmt3-L109R for split SUMO-ID, as used in the work by F. Watts group (Jongjitwimol J, Feng M, Zhou L, Wilkinson O, Small L, Baldock R, Taylor DL, Smith D, Bowler LD, Morley SJ, Watts FZ. The *S. pombe* translation initiation factor eIF4G is Sumoylated and associates with the SUMO protease Ulp2. *PLoS One*. 2014 doi:10.1371/journal.pone.0094182) for in vitro SUMOylation assay on purified proteins, is a foundation for a new, broad project. It will raise the need of establishment of a new Mass spectrometry analysis pipeline.

Identifying SUMOylation sites on newly identified targets is the exciting plan for future investigation and utilisation of Pmt3-L109R and Pmt3- Δ GG might deliver interesting data. However, optimisation of in vivo use of both alleles is not a trivial task and might require a lot of time and attempts.

Lack of direct SUMO sites on described proteins in this manuscript does not change the overall conclusion and we believe is largely out of scope for the purpose of this publication.

3. Lines 675–677 describe applying different false discovery rate (FDR) cutoffs at various

levels, but the rationale is unclear. For high-confidence identifications, a consistent 1% FDR at the peptide and protein levels is standard. The authors should clarify and justify their FDR criteria, and ideally use a consistent 1% FDR for proteins, peptides, and any modifications to ensure comparability.

The initial decision to raise the FDR was done to increase the coverage of common proteins between the two different SPLIT-ID analyses (WT May 2024 and SUMO-KallR March 2025). Our goal was to compare the initial WT samples with newly obtained *SUMO-KallR*, as requested during revision process. Of note, in our experience with the Ion Accounting (IA) algorithm for DIA data, oftentimes using a slightly higher protein FDR yields more true positive proteins than false positive ones. It is especially true for pull-down and cell fraction proteomics. We believe this might be a general characteristic of DIA data, where the precursor windows are much less discrete, than in a traditional DDA MS/MS windows, and the fact that already 1 peptide causes a protein hit (either target or decoy).

However, considering the best standards within the proteomics community we adhered to Reviewer's suggestions and applied a protein level-FDR of 1% to whole dataset (WT and *SUMO-KallR* obtained in March 2025), which resulted in peptide level FDR ~0.4%.

The Ion Accounting (IA) algorithm works by applying the FDR cutoff at protein level. A search of a decoy databank containing randomized sequences derived from the target databank is performed in memory in-parallel, and target db proteins are reported until the set threshold in the decoy db is hit. Unfortunately, there is no way to directly control the peptide-level FDR when using IA, hence what we do is estimate it based on a retrospective target:decoy search performed on a databank containing target and randomized decoy sequences. We then take all peptide-level identifications from this search and calculate the percentage of decoy peptides identified relative to the target peptides, and report this as peptide-level FDR of our data.

We now include the IA peptide output files as well as Python scripts used for this procedure in the repository. Proper changes were made in the methodological section, and repository files are updated.

4. The statistical analysis reports p-values < 0.05 for significance, but no multiple hypothesis testing correction is mentioned. Given the large number of proteins tested, an appropriate correction (e.g. permutation-based FDR, Benjamini-Hochberg) should be applied to control false positives. The authors should specify how p-values were calculated and adjust for multiple comparisons.

We thank the Reviewer for this remark. The QiP documentation is unfortunately quite scarce on certain essential details of the statistical analysis. We, therefore, as mentioned earlier (point #1), turned to the Perseus software for a more transparent statistical analysis of the SPLIT-ID data in the revised manuscript. To test for significant differences, we used a two-sided t-test without a multiple hypothesis testing correction, but with a fold-change (difference) threshold. This is exactly as published in the original manuscript on SUMO-ID technique (Barroso-Gomila, O., Trulsson, F., Muratore, V. *et al.* Identification of proximal SUMO-dependent interactors using SUMO-ID. *Nat Commun* **12**, 6671 (2021); doi.org/10.1038/s41467-021-26807-6).

We are fully aware of the multiple testing correction concept, however we believe that it is too stringent for significance determination in quantitative analyses of proximity-labelling (PL) pull-down samples, especially for the SPLIT-ID technique. Due to the nature of the experiment, such samples are characterized by a much higher overall variability, than that of

standard whole cell proteomics, where the majority of signals is much more consistent. Hence, an overly stringent statistical determination might result in true positive changes being rejected.

In the current version of the datasets there are 108 and 26 changes indicated solely by the p-value significance ($p \leq 0.05$) for WT and KallR analyses, respectively. The t-tests were performed on 638 and 621 protein observations, which means that out these changes 32 and 31 could occur just by chance for WT and SUMO-KallR analyses, respectively. For the WT analysis, the count of 32 is well below 108, which means that many of the changes could be in fact true positives. On the other hand, if we apply a Benjamin-Hochberg FDR correction the lowest q-value obtained reaches only down to 0.14, and there are only 9 proteins with a q-value below 0.3. Furthermore, q-values for bait proteins (which are highly expected to be truly enriched) are 0.17 and 0.31 for Pmt3 and Rad52, respectively. This, in our opinion, shows that the multiple hypotheses testing correction will lead to receiving much less informative, perhaps even uninformative, data due to excessive stringency. Noteworthy, we do not rely on p-value alone, but also on the magnitude of the change (fold change) in assuming our significance thresholds. This in the end leaves 96 changes, out which 70 are enrichments, a number quite close to 76 (number resulting from subtracting the number of “by-chance” changes at $p\text{-value} \leq 0.05$ from all p-value indicated changes). For the SUMO-KallR analysis, the count of 26 is near 31, indicating many of the changes may be indeed false positives. This is however still in-line with one of our final conclusions from the Split-SUMO-ID experiments, that loss of SUMO chains diminishes SUMO-dependent interactions at the Rad52 repair sites. Majority of changes observed in the SUMO-KallR analyses could be in fact “by-chance” changes, as the reproducibility analyses show that the SUMO-KallR-ID samples are in fact much alike to their controls, and no clear clustering can be really determined. Nevertheless, Pmt3 and Rad52 are still significantly enriched in the SUMO-KallR-ID relative to controls (which we would still argue as highly expected). This however stands true only if no multiple hypothesis testing correction is performed, otherwise the q-values are 0.68 and 0.84 for Pmt3 and Rad52, respectively. This further solidifies our opinion that multiple hypothesis testing correction should not be applied to the datasets at hand for significance determination, as it is simply too stringent for this type of data.

All above mentioned statistical workflow steps can be tracked in the Perseus project file (250331_CBNDK_KKR_SPLIT_ID.sps) uploaded in the repository.

To sum up, we have performed our SUMO-ID analysis on three independent biological replicates, utilized FDR set at 1% according to the remarks of the Reviewer and adopted the statistical testing similarly as in original manuscript on SUMO-ID (Barroso-Gomila et al. 2021).

5. Four SUMOylation sites on Rad52 have been reported in *S. cerevisiae* (two in the N-terminal region, one in the central domain, one in the C-terminal region). The exact number of SUMO moieties in the polySUMO chain is not known. In Figure 7a, the authors fuse four SUMOs to the C-terminus of Rad52, which appears to mimic the four monoSUMOylation sites. However, since the reported SUMOylation sites are primarily located near the N-terminus, the authors should elaborate on the rationale for choosing a C-terminal fusion. If the goal is to simulate endogenous SUMOylation, it may be more appropriate to fuse SUMOs to the N-terminus. The rationale for using a 4×SUMO fusion other than 3 or 5 should also be explained, and an N-terminal fusion should be tested to confirm that the construct functions similarly.

S. pombe Rad52, formerly known as Rad22, exhibits little homology with *S. cerevisiae* Rad52 protein, although it fulfil an analogous role in homologous recombination and single strand annealing. A report from Kohli lab shows that N-terminus is not conserved between scRad52 and spRad52 (Octobre G, Lorenz A, Loidl J, Kohli J. The Rad52 homologs Rad22 and Rti1 of *Schizosaccharomyces pombe* are not essential for meiotic interhomolog recombination, but are required for meiotic intrachromosomal recombination and mating-type-related DNA repair. *Genetics*. 2008; doi: 10.1534/genetics.107.085696). See below the figure from this work, underlining the differences between two orthologues.

REDACTED

Of note, spRad52 does not contain analogous SUMOylation sites at the very N-terminus as scRad52 posses. Nevertheless, a report by Ho et al 2001 (Ho JC, Warr NJ, Shimizu H, Watts FZ. SUMO modification of Rad22, the *Schizosaccharomyces pombe* homologue of the recombination protein Rad52. *Nucleic Acids Res*. 2001 doi: 10.1093/nar/29.20.4179) showed SUMOylation of Rad52 (Rad22) and our new data generated during revision process confirms it (Fig. 7b-c).

We have performed a bioinformatics analysis of spRad52 sequence using GPS SUMO 2.0 and DeepSUMO tools. Both software indicated potential SUMOylation sites at the very C-terminus (sequence DATVDKKAKKG*). K36 and K108 at the N-terminus were also marked as potential modification site. As presented below, there's also accumulation of potential SUMOylation sites within the core of the protein (K135, 185, 197 and 215). Moreover, at least four putative SIMs were marked within the sequence of spRad52.

In total, there are 35 lysines in *S. pombe* Rad52, and both bioinformatics analyses suggested that potential SUMOylation sites are located within the core and at C-terminus of the protein, in contrast to *S. cerevisiae* ortholog.

We therefore decided to prepare a construct with linear SUMO chain, placed at the C-terminus of the spRad52. Western blot clearly shows that complete linear chain is attached to Rad52 based on the molecular mass of fusion protein.

The estimation about the length of SUMO chain was done based on the report Sriramachandran et al 2019, where the authors tested artificial polySUMO targets in budding yeast heterologously expressing RNF4 or ARCADIA STUBLs (Sriramachandran, A.M., Meyer-Teschendorf, K., Pabst, S. *et al.* Arkadia/RNF111 is a SUMO-targeted ubiquitin ligase with preference for substrates marked with SUMO1-capped SUMO2/3 chain. *Nat Commun* **10**, 3678 (2019); doi.org/10.1038/s41467-019-11549-3).

In that manuscript, a linear chain, consisting of 4 SUMO units was sufficient for triggering STUbL activity, as efficiently as native lysine-linked SUMO-2 chain. We therefore chose to tag endogenous Rad52 with 4 active SUMO (Pmt3) particles. Not only it allows to detect heavier form of Rad52 as the sole version of Rad52 in relevant Rad52-Schain mutants, but also we have shown that it can undergo further SUMO attachment, presenting similar profile to polySUMOylation on control Rad52-6His-3Flag on Ni-NTA pulldown (Fig. 7b). We also showed several in vivo phenotypes of Rad52 with linear SUMO chain (Rad52-Schain) as described in the manuscript. We believe, since in our hands 4 SUMO units are sufficient to trigger global reduction of rad52 and decrease of centromeric recombination, that Rad52-Schain behave indeed as an artificial, linear SUMO chain. Given above, although we agree that testing N-terminal fusion of a SUMO-chain to Rad52 would be interesting for additional insight into features of SUMO chains function, we still believe that for the purpose of this manuscript it won't change presented conclusion and thus is out of scope.

6. The western blot in Figure 7c intended to validate the Rad52-4SUMO fusion is unclear. Additional controls are needed. For example, the authors could express Rad52- HF in wild-type, Pmt3SUMO, and Pmt3-KallR backgrounds, perform Ni-NTA pulldown, and probe the blots with both anti-Flag and anti-SUMO antibodies. Running the Rad52-4SUMO-HF sample alongside these controls on the same blot would allow direct comparison of band patterns and confirm the presence of polySUMOylated Rad52 species.

We thank the Reviewer for raising this issue. We have now performed a complete experiment with relevant controls to clearly present the SUMOylation patterns of unmodified Rad52 and fusion protein Rad52-Schain on anti-Flag and anti-SUMO Western blots (Fig. 7b). It confirmed that linear SUMO chain attached to Rad52 is stably expressed and might undergo further modification, similar to endogenous Rad52-HF polySUMOylation.

7. Given existing knowledge of SUMOylation in yeast proteins, the authors should identify which lysines are SUMOylated in their system. This could be done by site-directed mutagenesis of candidate lysine residues (followed by western blot to test loss of SUMOylation) or by MS analysis using the Pmt3-L109R strain to detect di-glycine remnants on modified peptides. Mapping the SUMO sites on Rad52 (and other key proteins) would strengthen the mechanistic conclusions.

The point raised by the Reviewer is an interesting and exciting research direction. However, we need to emphasise, that the aim of our work was to study the unexplored effects of loss of polySUMOylation by investigating *SUMO-KallR* allele. Another feature of our work is the reconstitution of artificial SUMO chain on a single protein (Rad52-Schain in *SUMO-KallR* background). Given the multiple, potential SUMOylation sites within Rad52 and Pcn1, it is not a trivial task to mutate them and subsequently present solid evidence on the overall effects of loss of SUMOylation on *S. pombe* DNA metabolism. Thus, this constitutes an independent, new direction of research, in frame of next scientific project due to financial constraints. We believe, that exact description of SUMO sites in Rad52 or Pcn1 is falling out of scope for presented manuscript and lack of this information does not alter our proposed model and conclusions.

8. While Pli1 is identified as mediating PCNA polySUMOylation, it is important to determine which SUMO E3 ligase is responsible for Rad52 polySUMOylation. The authors should use a similar approach (e.g. testing Rad52 SUMOylation in mutants of known SUMO E3s) to identify the ligase(s) involved. This would add mechanistic detail to the model of how SUMO chains on Rad52 are generated.

We have performed a Ni-NTA pulldown of endogenous Rad52-6His-3Flag from WT, *pli1Δ* and truncated protein *nse2RINGΔ* with deleted last 56 amino acids, including the C195 and H197 conserved residues, thus inactive as E3 SUMO ligase. We clearly present that polySUMOylation of Rad52-HF is conducted by Pli1 in vivo. In *S. pombe* Pli1 (*scSiz1* and *Siz2*) is the sole E3 SUMO ligase responsible for polySUMOylation.

9. The authors observe a strong decrease in proteins identified by Rad52-SUMO-ID in the Pmt3-KallR strain (lines 351–354), which contrasts with a prior Molecular Cell study. A plausible explanation is that without SUMO chains on Rad52, many SIM-dependent interactors no longer bind. This implies that several hits in the SUMO-ID may be proteins that interact via SUMO–SIM binding, rather than being PolySUMOylated themselves. The authors should discuss this possibility. It could mean that some identified “interactors” in the wild-type strain are indirect (SIM-binding), and their loss in the KallR strain explains the reduced list. Explicitly considering which hits are truly SUMOylated versus SUMO-binding is important.

We thank the Reviewer for this point. We discussed this issue more broadly, as SIM-mediated interactions were not considered by us as an explanation for this surprising result. This is a very important idea, that is now included within the model and is broadly discussed in the Discussion section. We cited the work from Thon lab (Kohler et al 2015) where a SUMOylome analysis was presented. It shows some of our detected proteins were already described as SUMOylation targets (Pol3, Htb1, Rvb2). Pcn1's SUMOylation was so far not published and we extensively studied it in our work (see also the answer to point 4 of Reviewer 3). Thus, we have adjusted the conclusions in our work and rewritten the Discussion section according to the comment raised by the Reviewer.

10. The schematic model in Figure 8 should be clarified. Currently, it is not clear which additional site of PCNA and site of Rad52 are SUMOylated on each side of the model. The authors should explicitly mark the SUMOylated substrates in the diagram. In particular, including the roles of Cnp1 and Cnp3 (and their SUMOylation status, if known) would provide a more complete picture of the centromere maintenance mechanism. Additional annotation or explanation in the figure legend might help convey how these factors interact in the proposed model.

In the revised version, we present an adjusted model, according to the comments raised by the Reviewer. We have changed the representation of SUMOylation, as explicitly explained in the previous points. We clearly marked the Pli1-dependent polySUMOylation and the effect of SUMO chains loss at the centromeric region. We have now also placed Cnp1 and Cnp3 within the model, as suggested by the Reviewer.

Minor Points

1. In lines 23–25 the authors refer to “site-specific SUMOylation,” but only the single known site K164 of PCNA is mutated in the study. This wording is misleading unless additional SUMOylation sites have been systematically identified. The authors should clarify or revise the phrasing (e.g. specify “SUMOylation at PCNA-K164” or remove “site-specific” if only one site is analyzed).

We thank the Reviewer for raising this point. In the revised version, we have introduced following changes to the abstract:

“To investigate SUMO-dependent interactome at the sites of Rad52 repair, we used split-SUMO-ID proteomics approach. It allowed analysis of local SUMOylation content at the Rad52 repair sites, and enabled identification of the essential replication factor PCNA.”

2. In lines 343–344, the term “SUMOylome” is used. This suggests a comprehensive list of SUMOylated proteins, but the Split-SUMO-ID dataset likely includes proteins that bind via SIMs and are not directly SUMOylated. To avoid confusion, a more precise term like “SUMO-dependent interactome” or an explicit explanation would be better.

We are grateful for this point. We have now changed “SUMOylome” and refer as SUMO-dependent interactome at Rad52 repair centers.

We are grateful to both Reviewers for their thorough and insightful evaluations of our manuscript. The comments raised by Reviewers have drawn our attention to important aspects that were not sufficiently addressed in the original version, particularly regarding the more mechanistic impact of polySUMOylation on the replication stress response.

Based on the comments and criticism of both Reviewers we have substantially revised the manuscript. This includes major rewriting of the text, redesigning several experiments, and adding new data and analyses. Before addressing each comment point-by-point we would like to summarize the key revisions made:

- 1) We have added new experiments that deepen our understanding of the role of SUMO chains in the maintenance of centromere stability in untreated conditions.
- 2) We have removed the figure related to rDNA maintenance. While it contained important findings (including bidimensional gel electrophoresis of rDNA locus in unstressed conditions), we decided that it would be better suited for a separate manuscript following additional experiments, and set the focus of current research exclusively on centromeres. This allowed us to elucidate the role of SUMO chains in the maintenance of centromeres in fission yeast.
- 3) We extended the SUMO-ID analysis to include SUMO-KallR background and found that polySUMOylation occurs frequently at repair sites.
- 4) We thoroughly examined and functionally tested most interesting protein of the identified hits – DNA replication factor PCNA. We showed its SUMOylation status in *S. pombe*. We also generated and characterized an artificial Rad52-SUMOchain fusion protein.
- 5) We performed additional in vivo experiments, including Ni-NTA pulldown of overexpressed PCNA-6His-3Flag and PCNA-K164R-6his-3Flag across genetic backgrounds to disseminate the repair pathways employed at suspected replication arrest sites within centromeres.
- 6) We have refined the model, to incorporate the new findings, proposing a role for SUMO chains in channelling the repair at stalled forks into Rad8 (scRad5)-dependent pathways. We also showed that both PCNA and Rad52 are targets of polySUMO attachment that control centromeric rearrangements.

We sincerely hope that the Reviewers will find the revised manuscript significantly improved and that the additional data address their concerns.

Please find below the point-by-point responses to specific comments raised by both Reviewers.

Reviewer #2 (Remarks to the Author)

In this manuscript, Markowska et al. investigated the impact of loss of SUMO chains on DNA replication at rDNA and centromere regions in fission yeast *S. pombe*. One interesting observation is that cells deficient in SUMO chain formation exhibit elevated replication stress and DNA damage. Further evidence indicate that SUMO chain formation is required for rDNA stability and the precise centromere localization of CENP-A protein. In addition, SUMO chain-deficient cells show increased replication fork stalling at rDNA region as well as enhanced recombination rate at centromeres. Finally, a split-SUMO-ID proteomic approach was used to identify Rad52-SUMO associated proteins. Although the results from this research indicate the role of polySUMOylation in the replication of the difficult-to-replicate loci

(rDNA and centromeres), this manuscript lacks sufficient mechanistic details about the precise role of polySUMOylation in DNA replication at these regions. Major revisions are required before this manuscript can be considered for publication by Nature Communication.

We thank the Reviewer for the thoughtful comments and constructive suggestions. In response, we have undertaken a major revision of the manuscript, in accordance with criticism below. We have rewritten initial manuscript, significantly modified the manuscript, and added several new data (New Figures 5, 6, and 7). We set the focus of revised manuscript on centromere maintenance and given the volume of new data, we have decided to remove rDNA-related section from the revised manuscript. We believe this improves the clarity and flow of the revised manuscript. We hope that Reviewer finds the revised version significantly strengthened and proposed mechanism behind the role of SUMO chains in fine tuning centromere stability solid and worth publishing.

Major points

1. Despite the interesting observations of increased RPA or Rad51/2 foci in rDNA or centromere regions in cells lacking SUMO chains are very interesting, the results in this manuscript are not sufficient to demonstrate how SUMO chains specifically affect replication or nucleosome organization of these repetitive sequences. It is unclear if polySUMOylation functions during normal DNA replication process in these regions or only when DNA damage/replication stress occurs.

In the revised version of manuscript we showed by EdU incorporation assay that without SUMO chains cells suffer from exacerbated endogenous replication stress (Figure 1f-g). Furthermore, our new data regarding PCNA modifications suggest that during unchallenged conditions, SUMO chains contribute to balanced activation of post replication repair within centromeres (New Figures 5 and 6). In the absence of SUMO chains PCNA undergoes excessive polyubiquitination, leading to increased centromeric recombination (New Figure 5b, 6b-c, 6e). Thus, we propose that polySUMOylation functions during normal DNA replication, and when replication forks are impeded by natural obstacles (i.e. repetitive regions like centromeres) polySUMOylation safeguards balanced activation of salvage pathways.

2. Previous studies in other organisms demonstrate the SUMOylation of proteins associated with rDNA and centromeres. Identification of the SUMO proteins that directly contribute to rDNA stability and Cnp1/CENP-A centromere localization will strengthen this work.

In the revised manuscript, the focus of investigation was set exclusively on centromeres. We attempted to perform the SUMO-ID using the centromeric protein Mis6. However we failed to obtain viable Mis6-CTurbo-12Pk strain. On the other hand, based on the SUMO-ID data generated for Rad52, we have identified PCNA as a SUMO target (Figure 4c). We found that abrogation of PRR by introduction *pcn1-K164R* mutation or *rad8+* (scRad5) deletion into *SUMO-KallR* background suppressed its sensitivity to thiabendazole (TBZ) (New Figure 6a). In the revised manuscript we have explored PCNA modifications and their involvement in the repair pathways choice at replication forks arrested within centromeres. We showed that PCNA undergoes SUMOylation and upon SUMO chain loss it became excessively ubiquitinated in a manner dependent on Rad8 (New Figure 6c). We also measured centromeric recombination and found that abolishment of post replication repair (PRR) by introduction of *pcn1-K164R* allele into *SUMO-KallR* mutant is reducing increased centromeric recombination rates (New Figure 6e). However, we found that aberrant Cnp1 spreading beyond centromere core seems to be not related to excessive recombination at centromeres (New Figure 6f). We also have developed a novel

plasmid construct for tagging protein of interest with linear four SUMO-GG units (truncated, active protein with exposed diglycine motif) derived from pFA6a-KanMX. This approach enabled us to generate a Rad52-SUMO chain fusion protein expressed from endogenous promoter and directly test the impact of polySUMOylation on Rad52 function (New Figure 7).

3. Recent evidence indicates the role of the polySUMOylation in protein delocalization, and the polySUMO-triggered STUbL and Cdc48 complex are required for this delocalization. One possibility is that the lack of SUMO chain in SUMO-KallR mutants contributes to the rDNA stability and aberrant Cnp1 centromere loading through the polySUMO pathway, but this possibility was not tested in this research.

It was proposed that Cdc48 contributes to incorporation of Cnp1 into centromeric chromatin, specifically within the core and to some extent imr region, but not otr sequences. (Nakase Y et al. 2024 doi: 10.1242/bio.060287. 2024).

Notably, we identified Cdc48 with SUMO-ID assay along with several proteasomal components, supporting the link between SUMOylation and proteasome-mediated turnover of repair complexes. Interestingly, majority of positive hits from SUMO-ID detected in proximity of Rad52 were lost, when the only SUMO form was SUMO-KallR. These findings are consistent with the canonical role of SUMO chains in targeting proteins for degradation.

To further investigate this, we used *slx8-29*, a temperature-sensitive allele that inactivates SUMO-targeted ubiquitin ligase in *S. pombe* (data presented on New Supplementary Figure 6). We validated the allele by introducing *SUMO-KallR* mutation and confirmed by Western blot that it leads to a decrease in SUMO conjugates accumulation in *slx8-29 SUMO-KallR* (New Supplementary Figure S6a). We then examined Cnp1 localization at both permissive and restrictive temperatures. The *slx8-29* mutation alone caused aberrant Cnp1 distribution, only at restrictive temperature. SUMO-KallR suppressed several phenotypes of *slx8-29* at restrictive temperature, upon Slx8 depletion, however it did not reduce spreading of Cnp1. These results suggest that the spreading of Cnp1 might be both dependent on attachment of polySUMOylation and STUbL-mediated protein turnover pathway.

4. Fig. 2h shows efficient Chk1 phosphorylation in SUMO-KallR mutant cells treated with MMS, but it could be more informative to follow Chk1 dephosphorylation kinetics during the recovery from MMS treatment. It is possible that polySUMOylation facilitates checkpoint silencing by disassembling the protein complex at DNA damage foci.

We performed complementary experiments (New Supplementary Figure S3e) under the same conditions as those shown on Supplementary Figure 3c. However, we did not observe any clear differences between WT and *SUMO-KallR* mutants. During the course of these experiments, Chk1 remained phosphorylated, making it technically challenging to accurately determine the timing of its dephosphorylation simply by extending the experimental time window. We hope the Reviewer agrees that, while informative, this aspect does not represent a central point of the current study and does not impact the main conclusions of the manuscript.

5. It seems that SUMO-KallR mutant cells exhibit increased Cnp1 chromosome association throughout the genome, even though the increase is more dramatic at centromere regions (Fig. 5). Recent studies

from the Basrai lab in budding yeast indicate that STUbL Slx5 prevents mislocalization of centromeric CENP-A (Cse4) to chromosome arms by promoting Cse4 degradation. This possibility should be examined in *S. pombe* SUMO-KallR mutants.

To address this point we have recalculated ChIP-seq data, to more rigorously remove the background noise. The signal of CFP-Cnp1 in *SUMO-KallR* mutant (blue) was still greater than WT (red) and both are solely increased in the region of centromere. The *SUMO-KallR* and WT signal is specific to centromere and throughout the chromosome overlaps with green line (untagged control). We now present in the New Supplementary Figure 5c-e the whole chromosome ChIP-seq profile. Below a detailed description of action undertaken.

To specifically examine whether the peaks are present only at centromeres or also along the chromosomes, we performed an additional advanced ChIP-seq data analysis, including a more rigorous removal of background noise. Using MACS3, we computed the local bias by taking the maximum bias from surrounding 1kb, 10kb, the size of fragment length *d*, and the whole genome background. Finally, we combined and generated the maximum background noise, scaled the ChIP and control to the same sequencing depth, called peaks on score track using a cutoff, and then generated plots.

More rigorous background noise removal allowed us to conclude that the peaks are present only at centromeres. We believe that the very small SUMO-KallR peaks seen along the chromosomes are not meaningful, because they are the same height as the Untagged peaks, indicating unspecific binding of antibodies. We have replaced the centromere plots in Figure 2g (they are not significantly different from those in the original manuscript except for more rigorous background removal and alignment of the lines to 0 on the y-axis). We generated additional plots for whole chromosomes and included them in the Supplementary Figure 5c-e. Materials and methods are supplemented with relevant information regarding the analysis of sequencing data. The results of ChIP-seq analysis have been described in greater details and discussed in the appropriate sections of the manuscript. The Bigwig files have been replaced in the GEO database with appropriate annotation.

Considering the work from Basrai lab, we have performed cycloheximide chase experiment (New Supplementary Figure 6c), however Cnp1 was surprisingly stable, which is distinct when compared to budding yeast Cse4 ortholog. It seems that Cnp1 turnover is limited in *S. pombe* and neither STUbL mutation or *SUMO-KallR* allele does not clearly leads to stabilisation of the level of Cnp1 during CHX time course. The profile of Cnp1 from *SUMO-KallR* resembled WT samples. Thus, we concluded that STUbL is not directly contributing to Cnp1 turnover, presumably it impacts the level of Cnp1 by targeting kinetochore-associated proteins that contribute to Cnp1 incorporation into the centromeric chromatin.

6. The authors used Turbo-ID to detect proteins that associate with Rad52-SUMO. While it is a useful tool to study transient protein interactions, the results do not provide sufficient insight into the precise role of Rad52 SUMOylation in the altered rDNA and centromere organization.

To strengthen the SUMO-ID part, we have added several new experiments. Specifically, we performed SUMO-ID using NTurbo-*SUMO-KallR* allele in *SUMO-KallR* background. Deconvolution and searching of raw MS data for both analyses (the original SUMO-WT-ID and new SUMO-KallR-ID) was unified to follow the same parameters, as described in detail in the updated Materials and Methods section. Proper corrections are marked in the Materials and Methods section. The PRIDE repository dataset related to

this study was also updated. More detailed description of action taken and the precise impact on final datasets are included in response to Reviewer 3.

Pcn1 was among identified hits in SUMO-ID. We examined in details PCNA SUMOylation status (earlier unknown in *S. pombe*). We expressed Pcn1-6his-3Flag and optimised Ni-NTA pulldown (New Figure 5 and 6). By that approach we were able to asses that PCNA undergo SUMO chain modification in Pli1 dependent manner. PolySUMOylation on PCNA restricts its excessive polyubiquitination. We also present several lines of genetic evidence, including suppression of TBZ sensitivity of SUMO-KallR by introducing *pcn1-K164R* or *rad8+* deletion.

Additionally, we have generated a Rad52-SUMOchain fusion protein expressed from endogenous promoter and directly test the impact of polySUMOylation on Rad52 function (New Figure 7). Functionally, we demonstrated that Rad52-SUMOchain has expected signal on anti-SUMO western blot after Ni-NTA pulldown (New Figure 7c), and can partially suppress TBZ sensitivity of *SUMO-KallR* mutant (New Figure 7d). Importantly, SUMO-chain modified Rad52 no longer promotes increased centromeric recombination (New Figure 7e) and overall reduce ability to form Rad52 foci examined by immunofluorescence. These results help clarify the functional significance of SUMOylation and SUMO chains in the maintenance of centromere stability.

Minor points

- Fig. 1b. The decrease of HMW SUMO proteins in SUMO-KallR mutant cells is not very clear. Quantification might help.

Depending on growth condition (reach media vs minimal media) the profile of SUMOylated proteins in *SUMO-KallR* mutant may vary. To better represent the expected reduction in HMW SUMO conjugates, we have provided a different exposure that more straightforward underly the difference between WT and SUMO-KallR strain (Supplementary Figure S1b). In the revised manuscript, on New Supplementary Figure 6a, a Western blot shows that *SUMO-KallR* mutant exhibit reduction in overall SUMOylated proteins both at 25°C and 34°C when compared to WT and is able to suppress accumulation of high molecular weight SUMO conjugates resulting from dysfunctional STUbL, as expected based on previous reports.

- The MW markers are missing in all WB results.

Corrected.

- Several figures include microscopy images for the SUMO-KallR mutant but lack images for the control used. This begins with Figure 1G but is seen throughout the figures.

All controls added.

- Line 97-99: In addition to the possibility of limited recruitment of HR factors in SUMO-KallR mutants, compromised disassembly/delocalization is also possible.

Rephrased based on new data as follows: "Our work provides evidence that SUMO chains are essential for resolution of spontaneous replication stress at centromeres. "

- Line 107: Synthetic sickness? It seems that single SUMO deletion mutant exhibits slow growth, thus this phenotype is not synthetic sickness.

pmt3 Δ remains viable in contrast to smt3 Δ from budding yeast, however exhibits an extreme slow growth phenotype, followed by viability loss. The statement 'synthetic sickness' in regard of pmt3 Δ is removed from revised manuscript.

- Fig. 1e/line 120: "...indicating increased spontaneous DNA damage". Another explanation is compromised disassembly of protein complexes at DNA damage foci.

Another explanation added.

- Fig. 4a/b: The possibility of a kinetochore clustering defect in SUMO-KallR mutant cells needs to be considered.

The declustering of centromeres in *SUMO-KallR* was slightly increased, although not significantly enhanced compared to WT (Supplementary Figure 11e).

- Fig. 6e: The authors claim that mutant rad51-3A, showed the most severe sensitivity to TBZ. This is confusing as the most severe defects were seen in rad51 Δ SUMO-KallR and rad52 Δ SUMO-KallR double mutants. The authors should change the wording for clarity.

Text corrected as follows: "Loss of SUMO chains also enhanced TBZ sensitivity in *SUMO-KallR rad51 Δ* and *SUMO-KallR rad52 Δ* double mutants compared to single mutants (Fig. 3c). It indicates that HR and SUMO chains might contribute to centromere stability at least partially through independent pathways. Interestingly, *SUMO-KallR* allele was epistatic to the *rad51-3A* separation-of-function mutant (defective in strand exchange, but proficient in DNA binding⁶), suggesting potential involvement of polySUMOylation in regulation of Rad51 catalytic activity in the context of centromere maintenance (Fig. 3c)."

- Fig. 7b: There is a concern with the anti-HA blot as the streptavidin pulldown of biotinylated proteins shows that HA-SUMO can be non-specifically pulled down in the strain expressing only one subunit of TurboID (HA-NTurbo-SUMO).

SUMO-ID experiments are done in WT or SUMO-KallR background. Both backgrounds contain endogenous biotin ligase Bpl1, encoded by an essential gene. Bpl1 is presumably responsible for some biotinylation events that occur even in the control split-TurboID samples, when cultures are starved without biotin for 21 hours and then resupplemented with 50 μ M biotin for 3 hours. We hope Reviewer appreciates that the large and visible increase of the biotinylated proteins occurs exclusively in the samples co-expressing both subunits of TurboID.

- Fig. S4d: It appears that biotin itself can increase protein SUMOylation (WT + HA-NTurbo-SUMO), regardless of split-TurboID reconstitution. The authors should clarify this result. As stated above, the strains for SUMO-ID are grown without biotin, thus stop dividing after 21h of continuous growth. After addition of the biotin to media cells are reinitiating the growth and that leads

to the general increase in protein levels, especially of lower molecular size – compare Ponceau S pictures (Supplementary Figure S7e). On +biotin (right side), the intensity of protein level is increased compared to – biotin (left side Fig. S7e). On the other hand, stimulated growth by the addition of biotin to cells starved for 21h in media free of biotin might also impact SUMO attachment as a factor controlling cell metabolism.

Reviewer #3 (Remarks to the Author)

The manuscript entitled "SUMO chains fine-tune DNA replication and limit recombination at difficult-to-replicate sites" by Markowska and colleagues reports on the role of SUMO chains in maintaining genome integrity in fission yeast. To prevent the formation of SUMO chains the authors use an allele SUMO-KallR that can be conjugated to lysines but cannot itself be sumoylated. Cells that express SUMO-KallR exhibit a number of phenotypes including increased RPA, Rad51 and Rad52 foci, and increased instability of the rDNA loci, which is accompanied by replication stalling in the rDNA and Rad52 foci in the nucleolus. Furthermore, other hard-to-replicate loci such as the centromeres are also destabilized by the SUMO-KallR allele. To some extent the SUMO-KallR appears to be a hypomorph with an intermediate phenotype of WT and SUMO Δ . Finally, the authors have developed a split-SUMO-ID method, which they use to identify protein in the proximity of Rad52, which are sumoylated or bind to sumoylated proteins. The manuscript is well-written and the data are of high quality, but unfortunately the study does not extend much from previous studies in terms of elucidating the specific role of SUMO chains and none of the hits from the split-SUMO-ID screen are validated and characterized. For specific suggestions and recommendations, see below:

We thank the Reviewer for the valuable suggestions and constructive criticism. In response, we have significantly modified the manuscript and added several new datasets. Given the main focus is set on centromere maintenance and the volume of data presented in this context, we have decided to remove rDNA-related section from the revised manuscript. We believe this improves the clarity and conclusions of our manuscript.

Major issues to address

1. Line 126: the authors cannot conclude from figure 1 that replication stress is reduced by SUMO-chains, because replication stress was not examined in figure 1.

We have redesigned the Figure 1 and added a new experiment: EdU incorporation assay, to directly assess replication delays. This assay revealed a twofold increase in the number of cells displaying EdU signals in the *SUMO-KallR* mutant compared to WT in G2 phase. In light of previous findings on Figure 1, we concluded that the loss of SUMO chain leads to spontaneous replication stress and associated DNA damage. Further experiments that uncover SUMOylation of PCNA and its impact on channelling of post-replication-repair toward template switch supports that loss of SUMO chains leads to replication stress.

2. Line 134: the altered progression through S phase after release from HU arrest is not convincing in Fig. S2c. The claim should be supported by additional data.

FACS experiment (Supplementary Figure S2e) was performed in parallel to PFGE analysis (currently Supplementary Figure S2b-c), to show the ability of analysed strains to complete replication and thus

conclude that the delays in doubling migrating DNA content stem from inability to resolve replication intermediates. Minimal interval due to technical limitations was set at 15 min, so it was possible to collect both samples for FACS and samples for chromosome extraction.

We have provided additional, separate biological repeat of WT and *SUMO-KallR* with 10 min interval (Supplementary Figure S2f), that clearly shows the delayed recovery from HU-mediated S-phase arrest. We hope this new data convincingly supports our conclusion regarding the impact of SUMO chain loss on replication dynamics.

3. Figure 3d reports the % colocalization rather than the percentage of cells with each type of foci, which could be misleading given that the number of cells with foci also changes.

This data was extracted from the revised version of manuscript and will be used for separate work, devoted exclusively rDNA. We are grateful for this comment and agreed that in original description it was misleading. Our goal was to present the % of Rad52 foci relative to nucleolus.

4. A recent study identified poly-sumoylation as a regulator of rDNA replication in budding yeast (Gutierrez-Morton et al., 2024). The potential link to the current study should be discussed.

We have rewritten Discussion considering the suggestions from both Reviewers. The manuscripts that referred to SUMO chains roles in rDNA, replication and recombination are now included in our revised manuscript and discussed in frame of our results as follows: "Initial findings from budding yeast suggested that SUMO chains are rather dispensable for stress response and undergo dynamic removal by Ulp2 SENP protease⁴⁹. Later reports from budding yeast proved that SUMO chains-deficiency in *smt3-alkR* led to spontaneous DNA damage, replication stress and distorted rDNA structure, visualized by fluorescent microscopy of rDNA markers^{50,51}. A recent work by Wang lab explored in details rDNA distortion upon SUMO chain loss and proved that polySUMOylation promotes the release of Tof2 from nucleolus for mitotic exit by extracting Tof2-SUMOchain for proteasomal degradation through Cdc48-STuBL pathway⁵²."

5. Sumoylation has been linked to liquid-liquid phase separation. Might the aberrant Cnp1 localization reflect a defect in LLPS at centromeres?

We have performed a pilot experiments with 1,6 hexanediol, as a drug disturbing LLPS. We did not observe large differences between WT and *SUMO-KallR*, but this preliminary experiment does not exclude the role of SUMO chains in LLPS, as SUMOylation was reported to mediate proper LLPS. Thus, we find LLPS as an important point for potential future investigations, yet it will require introducing and optimization of novel approaches. We hope however that the Reviewer will agree that addition of LLPS analyses is out of scope for the purpose of this manuscript in its current form.

6. The split-TurboID is not tied in well with the rest of the manuscript and no validation or characterization of hits is performed e.g. actin. Can the authors observe interactions that are specific for SUMO chains using 6HA-NTurbo-SUMO-KallR?

We have now greatly expanded the part of SUMO-ID. First, we have added 6-HA-NTurbo-SUMO-KallR and expressed it in *SUMO-KallR* strain with Rad52-CTurbo. It underlined that majority of interactions at Rad52 repair centres required SUMO chain formation.

Deconvolution and searching of raw MS data for both analyses (the original SUMO-WT-ID and new SUMO-KallR-ID) was unified to follow the same parameters, as described in detail in the updated Materials and Methods section. Proper corrections are marked in the Materials and Methods section. In details, during the unification of MS data deconvoluting and searching parameters between the two SUMO-ID analyses, we employed signal intensity normalization between samples within each analysis (SUMO-WT-ID and SUMO-KallR-ID). This procedure, carried out by the processing software (QiP for Proteomics), helps to mitigate potential global load differences between samples. We explained the basis of this operation in the updated Materials and Methods section. In the original analysis the volcano plot on Figure 7c was unnaturally skewed to the right, which resulted in observing 277 enriched proteins (out of 972 total proteins identified) — majority of them being unrelated to our study. We initially interpreted this as an effect of a robust biotinylation increase in the pREP41-6HA-Nturbo-SUMO + Rad52-12PK-CTurbo strain. However, when reanalyzing the data in conjunction with the new SUMO-KallR-ID analysis, we now believe this right-side skew may be indeed related to load differences. As there is no reliable way to assess protein concentration on beads (despite the samples being at the same volume and protein concentration, and being incubated with the same amount of resin in each replicate), signal normalization is crucial for obtaining realistic results with proximity labelling assays. In the unified SUMO-WT-ID analysis, with signal intensity normalization employed, the volcano plot shows bimodal distribution, and the amount of significantly enriched proteins is now 94 (out of 913 total proteins identified). Notably, many of the proteins related to our study remained in the enriched fraction, which means that the actual ratio of *proteins-of-interest* : *all enriched proteins* increased. This ensures that for proper proximity labelling MS data interpretation the currently chosen data treatment is the correct option.

The PRIDE repository dataset related to this study was updated. In the SUMO-WT-ID dataset, after parameter unification the changes of Sfr1, Cnd3 and Has1 levels have lost statistical significance and are not displayed on the volcano plot anymore. Alteration of applied search parameters resulted in loss of Nuf2 protein identification, however a different kinetochore component, the Mal2 protein was identified in the reanalysis. These two proteins are sequentially unrelated. Such occurrence is unfortunately not uncommon for proteomic software searches carried out on data-independent acquired (DIA) raw data.

Nevertheless, changes for Pcn1, Cdc48, proteasomal components, Sgo2 from kinetochore and rDNA Nop58 protein remained valid in the newly analysed WT SUMO-ID. Noteworthy, some of these proteins-of-interest are also identified in the SUMO-KallR-ID analysis, however they are not significantly enriched (New Figure 4c and 4d). This further fortifies our confidence in the essentiality of SUMO chains at Rad52 repair sites, as we were able to directly compare these commonly identified proteins across both experimental designs.

Since PCNA was one of the positive hits, we verified its SUMOylation. To do it, we expressed Pcn1-6his-3Flag and optimised Ni-NTA pulldown (New Figure 5 and 6). By that approach we were able to assess that PCNA undergoes SUMO chain modification in Pli1 dependent manner. PolySUMOylation on PCNA restricts its excessive polyubiquitination. We also present several lines of genetic evidence, including suppression of TBZ sensitivity of SUMO-KallR by introducing *pcn1-K164R* or *rad8+* deletion.

We have also developed a novel plasmid construct for tagging protein of interest with four SUMO-GG units (truncated, active protein with exposed diglycine motif) derived from pFA6a-KanMX. This approach enabled us to generate a stable Rad52-SUMOchain fusion protein (New Figure 7). We showed that Rad52-SUMOchain can suppress TBZ sensitivity of SUMO-KallR mutant, thus underly that SUMO chains positively contribute to the maintenance of centromere. Artificial SUMO-chain on Rad52 led to its displacement from centromeres. Importantly, SUMO-chain modified Rad52 no longer mediated centromeric recombination. These results help clarify the functional significance of SUMOylation and SUMO chains in the maintenance of centromere stability.

We hope that Reviewer will find split-TurboID experiments tied well with the rest of manuscript in the current form. We focused on Pcn1 for validation of SUMO-ID, and assessed SUMO chains impact on Rad52, as both proteins contributed to the elevated centromeric recombination measured in *SUMO-KallR* mutant.

7. Several studies have analyzed SUMO chains in budding yeast e.g. (Bylebyl et al., 2003; Newman et al., 2017; Psakhye et al., 2019). The results of the current study should be compared in greater detail to previous studies.

The Discussion section has been extensively rewritten to underly the substantial changes made throughout the manuscript. This includes the integration of new data, clarification of key findings, refinement of our model and comparison of our study with the previous studies.

Minor suggestions

1. Line 49: change “rearrangements” to “deletions”.

Sentence removed from the text as it was not relevant for Introduction.

2. The manuscript could benefit from a systematic correction of English grammar.

The manuscript was corrected.

3. Line 176: the authors should be more specific about the observed change to chromosome III. To me, the band simply looks more smeary.

In revised version of manuscript all rDNA section was removed and we do not discuss the alterations of chromosome III. rDNA maintenance will be the subject of another manuscript.

4. Line 259: replace “interacted” with “colocalized”.

Corrected. It now reads as follows: “Since MMS-induced Rad52 foci often colocalize with Sad1, a spindle pole body protein anchoring centromeres³⁸, we examined spontaneous Rad52-Sad1 colocalization. It was significantly more frequent in the *SUMO-KallR* mutant, supporting increased centromeric engagement of repair factors. (Fig. S5f).”

5. Line 265: it is unclear what comparison “most severe” refers to. The double mutants are more sensitive to TBZ than *rad51-3A*.

It was raised also by Reviewer 2, and in present form of manuscript the text is modified as follows: “Loss of SUMO chains also enhanced TBZ sensitivity in *SUMO-KallR rad51Δ* and *SUMO-KallR rad52Δ* double mutants compared to single mutants (Fig. 3c). It indicates that HR and SUMO chains might contribute to centromere stability at least partially through independent pathways. Interestingly, *SUMO-KallR* allele was epistatic to the *rad51-3A* separation-of-function mutant (defective in strand exchange, but proficient in DNA binding⁶), suggesting potential involvement of polySUMOylation in regulation of Rad51 catalytic activity in the context of centromere maintenance (Fig. 3c).”

6. It is not entirely clear from the methods and legend what is shown in figure 7C. If I understand correctly it is the ratio of biotinylated proteins in pREP41 or pREP41-6HA-NTurbo-SUMO strains expressing Rad52-12PK-CTurbo in untreated conditions.

We are sorry for the misleading description of technique. Biotinylated proteins from each strain (*pREP41-6HA-Nturbo-SUMO* (control1), *Rad52-12PK-CTurbo* (control2) or *pREP41-6HA-Nturbo-SUMO + Rad52-12PK-CTurbo* (SUMO-ID, test sample)) were purified on streptavidin beads, but without an elution step. The bead-bound proteins were then analyzed by mass spectrometry in a quantitative manner. Volcano plot on Figure 7c (New Figure 4c, 4d in the revised manuscript) depicts the quantitative distribution and statistical significance for change in amounts of identified proteins purified in the test samples, relative to their amounts in the controls. $\log_{10}(FC)$ is the base 10 logarithm of the fold change value in amount of a given protein in the test samples relative to the controls; $-\log_{10}(p)$ is the negative base 10 logarithm of the probability significance value (p, t-test) for the change in given protein amount. Volcano plots comprise a convenient way of illustrating level-change data in the function of its statistical significance, as statistically relevant changes in both directions (up or down) are oriented towards either the left-hand corner (significant down-changes) or the right-hand corner (significant up-changes).

So, to summarize and update in accordance with the revised version of the manuscript, volcano plots in Figure 4c and 4d show proteins purifying in an unchanged (gray), depleted (light blue) or enriched (light orange) manner in preparations from the SUMO-WT-ID samples or the SUMO-KallR-ID samples, in relation to their respective controls. In the present form we have provided a clearer description.

References:

Bylebyl, G.R., I. Belichenko, and E.S. Johnson. 2003. The SUMO isopeptidase Ulp2 prevents accumulation of SUMO chains in yeast. *J Biol Chem.* 278:44113-44120.

Gutierrez-Morton, E., C. Haluska, L. Collins, R. Rizkallah, R.J. Tomko, Jr., and Y. Wang. 2024. The polySUMOylation axis promotes nucleolar release of Tof2 for mitotic exit. *Cell reports.* 43:114492.

Newman, H.A., P.B. Meluh, J. Lu, J. Vidal, C. Carson, E. Lagesse, J.J. Gray, J.D. Boeke, and M.J. Matunis. 2017. A high throughput mutagenic analysis of yeast sumo structure and function. *PLoS Genet.* 13:e1006612.

Psakhye, I., F. Castellucci, and D. Branzei. 2019. SUMO-Chain-Regulated Proteasomal Degradation Timing Exemplified in DNA Replication Initiation. *Mol Cell.* 76:632-645 e636.

All suggested references have been included in the revised version of manuscript.

REVIEWERS' COMMENTS

Reviewer #3 (Remarks to the Author):

The revised manuscript NCOMMS-24-57672B by Markowska and colleagues has addressed all of my remaining questions.

We are grateful for evaluating our work and helpful suggestions that developed the manuscript during revision process.

Reviewer #4 (Remarks to the Author):

The authors have addressed most of the previous concerns in a satisfactory manner. Before the manuscript can be considered for publication, a few issues still require clarification:

1. Statistical threshold inconsistency (Lines 1007–1008): the manuscript states that significance thresholds were set at “ $p \leq 0.05$ and fold-change (FCH) ≥ 1.5 (corresponding to $-\log_{10}(p) = 1.25$, $\log_2(\text{FCH}) = 0.585$)”. However, $-\log_{10}(p = 0.05)$ equals 1.301, not 1.25. Applying the correct cutoff would exclude additional protein hits currently listed in the supplementary tables SplitID_SUMO_WT_volc and SplitID_SUMO_KallR_volc. Please correct this inconsistency and update the results accordingly.

We are very grateful for picking up this inconsistency. The significance cut-off p-value was rounded to the nearest hundredth, hence the exact value for the cut-off is less than or equal to 0.055. This in turn equals approximately 1.25 after $-\log_{10}$ transformation. This rounding was noted in the Supplementary Dataset file (within the information found in yellow text boxes of the SplitID_SUMO_WT_volc and SplitID_SUMO_KallR_volc sheets) and could have been tracked within the Perseus project file. However, it should have been also stated in the main body’s methodological section, which unfortunately escaped our attention, hence we are, once again, very grateful for the remark. In the revised manuscript we updated the text to clearly indicate the exact value of the p-value significance cut-off, and its rounding to the nearest hundredth.

It reads as follows now: “Changes in protein quantity were identified with a both-sided t-test, assuming $p < 0.05$ (rounded to nearest hundredth; exact cut-off is less than 0.055) and fold-change (FCH) ≥ 1.5 as significance thresholds (logarithmic values: $-\log_{10}(p) = 1.25$, $\log_2(\text{FCH}) = 0.585$).”

Applying the 0.05 exact cutoff would eventually eliminate Rad52 from significantly enriched proteins in WT SUMO-ID dataset, but as an integral part of the assay, we believe it should be retained, based on both Western blot controls and overall increase in biotinylation signal in samples co-expressing Rad52-Cturbo with Nturbo-Pmt3.

Given that it was reaching $p=0.051$ based on 3 replicates used, we decided to employ the rounded cut-off to include it among enriched proteins.

2. Discrepancy in Perseus output: the “+” hits in column C (“Student’s T-test Significant SplitID_Controls”) of the supplementary spreadsheets are not consistent with the results in the uploaded Perseus project file (250331_CBNDR_KKR_SPLIT_ID.sps). The authors should clarify this discrepancy and ensure that the reported thresholds and lists of significant hits are fully aligned with the original analysis files.

Upon this remark, we closely reviewed the files once more, but we found no such discrepancy. Columns A–Q of the SplitID_SUMO_WT_volc and SplitID_SUMO_KalIR_volc resemble Matrix 12 and Matrix 61 of the uploaded Perseus project file. We double checked the hits marked with a “+” in the “C: Student's T-test Significant SplitID_Controls” columns and found them to be fully identical. The only difference between the supplementary spreadsheets and the mentioned matrixes of the .sps lies in the categorical mark of the “C: Student's T-test Significant SplitID_Controls_Filter”. In the the .sps file the mark reads “+_Keep” for proteins with a significant p-value and above the fold-change cut-off, while in the supplementary files we changed this to either Enriched or Depleted (dependent on change direction) — simply to facilitate Viewer’s comprehension. This codification was explained in Alternative Name fields of both matrixes’ for workflow clarity. However, as per Reviewer’s suggestion, to align the lists with the original analysis files we now reverted this codification to “+_Keep” in the supplementary files. Proper explanations for the “+_Keep” mark were also added in the yellow text boxes of the respective sheets. The file in the repository was accordingly updated as well.